# Nivolumab plus platinum-doublet chemotherapy in treatment-naive patients with advanced grade 3 Neuroendocrine Neoplasms of gastroenteropancreatic or unknown origin: The multicenter phase 2 NICE-NEC trial (GETNE-T1913)

The prognosis of patients with advanced high-grade (G3) digestive neuroendocrine neoplasms (NENs) is rather poor. The addition of immune checkpoint inhibition to platinum-based chemotherapy may improve survival. NICE-NEC (NCT03980925) is a single-arm, phase II trial that recruited chemotherapy-naive, unresectable advanced or metastatic G3 NENs of gastroenteropancreatic (GEP) or unknown origin. Patients received nivolumab 360 mg intravenously (iv) on day 1, carboplatin AUC 5 iv on day 1, and etoposide 100 mg/m$^2$/d iv on days 1–3, every 3 weeks for up to six cycles, followed by nivolumab 480 mg every 4 weeks for up to 24 months, disease progression, death or unacceptable toxicity. The primary endpoint was the 12-month overall survival (OS) rate (H$_0$ 50%, H$_1$ 72%, β 80%, α 5%). Secondary endpoints were objective response rate (ORR), duration of response (DoR), progression-free survival (PFS), and safety. From 2019 to 2021, 37 patients were enrolled. The most common primary sites were the pancreas (37.8%), stomach (16.2%) and colon (10.8%). Twenty-five patients (67.6%) were poorly differentiated carcinomas (NECs) and/or had a Ki67 index >55%. The ORR was 56.8%. Median PFS was 5.7 months (95%CI: 5.1-9) and median OS 13.9 months (95%CI: 8.3-Not reached), with a 12-month OS rate of 54.1% (95%CI: 40.2-72.8) that did not meet the primary endpoint. However, 37.6% of patients were long-term survivors (>2 years). The safety profile was consistent with previous reports. There was one treatment-related death. Nivolumab plus platinum-based chemotherapy was associated with prolonged survival in over one-third of chemonaïve patients with G3 GEP-NENs, with a manageable safety profile.

High-grade neuroendocrine neoplasms (NENs) are rare and aggressive tumors with a very poor prognosis. Over 90% originate in the lung and only 3% are of gastroenteropancreatic (GEP) origin[1,2]. The current standard of care still follows the treatment paradigm of small-cell lung cancer (SCLC), a far more common G3 NEN, although emerging molecular and clinical data increasingly question this approach[3]. Platinum-based chemotherapy (cisplatin or carboplatin and etoposide [EP]) is the standard first-line treatment for these patients, with objective response rates (ORR) of ~ 30% and a median overall survival (OS) of ~ 11 months[1–4]. The development of more

✉ e-mail: rgcarbonero@gmail.com

effective treatment strategies is, therefore, highly needed for these patients.

The immune system plays a pivotal role in cancer prevention, development, and progression, and immune checkpoint inhibition (ICI) has changed the treatment paradigm of many cancer types over the past decade, including some high-grade NENs such as Merkel cell carcinoma (MCC)[5] or SCLC[6,7]. Nevertheless, the role of immunotherapy remains controversial in G3 digestive NENs. Single-agent ICI with PD-1 or PD-L1 blockade is mostly ineffective in molecularly unselected, heavily pretreated patients with advanced GEP neuroendocrine carcinomas (NECs) (ORR < 8%, progression-free survival [PFS] 1.8-4 months, OS 5.1-7 months), with the only exception of toripalimab, that reported an ORR of 20% in G3 NENs primarily of GEP origin[8]. ORR was significantly higher in patients with the high mutational burden (TMB-H) or PD-L1 expression ≥10%, and about one-third of responders harbored *ARID1A* mutations[8]. In line with these observations, pembrolizumab also demonstrated efficacy in patients with TMB-H NENs[9]. Of note, TMB and PD-L1 expression, and other associated features predictive of response to immunotherapy, such as tumor-infiltrating lymphocytes, are significantly higher in high-grade NENs[10-13].

Dual PD-1 and CTLA-4 blockade have demonstrated increased efficacy compared with single-agent ICI in several tumor types, although more modest activity in pretreated G3 GEP NENs[14,15]. However, despite this modest activity, a subset of patients with pretreated G3 GEP NENs may achieve long-term survival[14].

On the other hand, emerging evidence suggests that chemotherapy may be synergistic with ICI as it induces an immunogenic cell death that can prime antitumor immunity within an immunosuppressive microenvironment[16-18]. This has been demonstrated in chemotherapy-naive patients with advanced SCLC, as the addition of atezolizumab or durvalumab to platinum-based chemotherapy has demonstrated improved survival and is now the standard of care as first-line therapy[6,7].

Here, we show the results of a phase II clinical trial that evaluated the efficacy and safety of nivolumab and carboplatin-etoposide in 37 patients with chemonaïve unresectable advanced or metastatic G3 NENs of GEP or unknown origin. The primary endpoint (12-month OS rate) was not met, but the combination was associated with an ORR of 56.8% and prolonged survival (>2 years) in over one-third of patients with a manageable safety profile.

## Results

### Patients

Between 2019 and 2021, 38 patients were enrolled in the study. All patients received the study treatment and were evaluable for safety. Out of these, 37 (97.4%) met the eligibility criteria for study entry and were evaluated for efficacy. One patient (2.6%) with a non-neuroendocrine neoplasm was withdrawn from the study and was only considered for safety assessment.

The main patient characteristics are summarized in Table 1. The most common primary tumor sites were the pancreas (N = 15, 40.5%), stomach (N = 6, 16.2%), and colon (N = 4, 10.8%), followed by esophagus, small bowel and rectum (N = 2; 5.4% each). Six patients had unknown primaries (16.2%). The majority were poorly differentiated NECs (N = 25, 67.6%), had a Ki-67 index greater than 55% (N = 25, 67.6%), were stage IV at diagnosis (N = 35, 94.6%), had ≥ 2 metastatic sites (N = 27, 73%), an Eastern Cooperative Oncology Group (ECOG) performance status of 1-2 (N = 26, 70.3%) and elevated baseline chromogranin A (N = 27, 73%) or enolase (N = 21, 56.8%). PD-L1 expression could be assessed in 35 cases and was positive in two (5.4% overall, 5.7% of assessed cases), one per CPS (10%) and TPS (1%) and the other one only per TPS (1%). Microsatellite instability was present in 1 of 14 evaluable patients (2,7% overall, 7,1% of assessed patients).

**Table 1 | Baseline characteristics of the study population**

| Characteristics | | G3 NENs N = 37 | G3 NETs N = 12 | NECs N = 25 |
|---|---|---|---|---|
| Age, years | Median (range) | 61 (28–84) | 58 (38–78) | 61 (28–84) |
| Gender, n (%) | Male | 25 (67.6) | 9 (75.0) | 16 (64.0) |
| | Female | 12 (32.4) | 3 (25.0) | 9 (36.0) |
| ECOG PS, n (%) | 0 | 11 (29.7) | 3 (25.0) | 8 (32.0) |
| | 1 | 22 (59.5) | 9 (75.0) | 13 (52.0) |
| | 2 | 4 (10.8) | 0 (0.0) | 4 (16.0) |
| Stage at diagnosis, n (%) | I | 1 (2.7) | 1 (8.3) | 0 (0.0) |
| | III | 1 (2.7) | 0 (0.0) | 1 (4.0) |
| | IV | 35 (94.6) | 11 (91.7) | 24 (96.0) |
| Differentiation, n (%) | NET | 12 (32.4) | – | – |
| | NEC | 25 (67.6) | – | – |
| Ki 67, n (%) | 21–55% | 12 (32.4) | 8 (66.0) | 4 (16.0) |
| | >55% | 25 (67.6) | 4 (33.3) | 21 (84.0) |
| Primary site, n (%) | Esophageal | 2 (5.4) | 0 (0.0) | 2 (8.0) |
| | Gastric | 6 (16.2) | 1 (8.3) | 5 (20.0) |
| | Pancreatic | 15 (40.5) | 6 (50.0) | 9 (36.0) |
| | Colonic | 4 (10.8) | 1 (8.3) | 3 (12.0) |
| | Rectal | 2 (5.4) | 0 (0.0) | 2 (8.0) |
| | Small intestine | 2 (5.3) | 2 (16.7) | 0 (0.0) |
| | Unknown | 6 (16.2) | 2 (16.7) | 4 (16.0) |
| Metastatic sites number, n (%) | 1 | 10 (27.0) | 5 (41.7) | 5 (20.0) |
| | ≥2 | 27 (73.0) | 7 (58.3) | 20 (80.0) |
| Metastasis sites, n (%) | Liver | 31 (83.8) | 11 (91.7) | 20 (80.0) |
| | Lung | 9 (24.3) | 2 (16.7) | 7 (28.0) |
| | Lymph nodes | 18 (48.6) | 2 (16.7) | 16 (64.0) |
| | Bone | 10 (27.0) | 4 (33.3) | 6 (24.0) |
| Previous surgery, n (%) | Yes | 6 (16.2) | 4 (33.3) | 2 (8.0) |
| | No | 30 (81.1) | 8 (66.7) | 22 (88.0) |
| | Unknown | 1 (2.7) | 0 (0.0) | 1 (4.0) |
| CgA, n (%) | <2x ULN | 7 (18.9) | 0 (0.0) | 7 (28.0) |
| | ≥2x ULN | 27 (73.0) | 11 (91.7) | 16 (64.0) |
| | Unknown | 3 (8.1) | 1 (8.3) | 2 (8.0) |
| Enolase, n (%) | <2x ULN | 13 (35.1) | 4 (33.3) | 9 (36.0) |
| | ≥2x ULN | 21 (56.8) | 8 (66.7) | 13 (52.0) |
| | Unknown | 3 (8.1) | 0 (0.0) | 3 (12.0) |
| LDH, n (%) | <2x ULN | 9 (24.3) | 8 (66.7) | 20 (80.0) |
| | ≥2x ULN | 28 (75.7) | 4 (33.3) | 5 (20.0) |
| PD-L1; n (%) | Positive | 2 (5.4) | 2 (16.7) | 0 (0.0) |
| | Negative | 31 (83.8) | 10 (83.3) | 21 (84.0) |
| | Unknown | 4 (10.8) | 0 (0.0) | 4 (16.0) |
| MSI; n (%) | Positive | 1 (2.7) | 1 (8.3) | 0 (0.0) |
| | Negative | 13 (35.1) | 4 (33.3) | 9 (36.0) |
| | Unknown | 23 (34.3) | 7 (58.3) | 16 (64.0) |

*CGA* Chromogranin A, *ECOG PS* Eastern Cooperative Oncology Group Performance Status, *LDH* lactate dehydrogenase, *MSI* microsatellite instability, *NEC* neuroendocrine carcinoma, *NET* neuroendocrine tumor, *PD-L1* programmed death ligand 1.

All patients had discontinued the study treatment at the time of the final analysis, 2 (5.4%) due to treatment completion (2 years) as scheduled and 35 (94.6%) prematurely. The primary reason for study discontinuation was disease progression, which occurred in 29 patients (78.4%). Three patients (8.1%) died during treatment due to disease progression, and one (2.7%) experienced a treatment-related death. Two patients (5.4%) discontinued the study treatment at the investigator's discretion.

## Treatment efficacy

With a median follow-up of 29.8 months (range: 22.2–37.5+) in alive patients, the median OS was 13.9 months (95% CI: 8.3-NR) and the OS rates at 6, 12 and 24 months were 78.4% (95% CI: 66.2–92.8), 54.1% (95% CI: 40.2–72.8) and 37.6% (95% CI: 24.8–57.1), respectively (Fig. 1A). The primary endpoint (12-m OS rate) was thus not met. OS did not significantly differ by tumor differentiation or Ki-67 index, although the median OS was notably higher for G3 neuroendocrine tumors (NETs) versus NECs (23.3 vs. 13.9 months, HR: 1.37; 95% CI: 0.5–3.5), and for Ki-67 ≤ 55 versus >55% (17.6 vs. 11.7 months, HR: 0.97; 95% CI: 0.4–2.4) (Supplementary Fig. 1). The patient with MSI tumor had long survival, being alive at data cut-off with an OS of 33.6+ months. The patient with CPS/TPS PD-L1 positive tumor was alive and had a survival of 22.8 months, while the patient PD-L1 positive per TPS only died after 2.2 months due to PD.

Multivariable analysis showed that tumor location was significantly associated with survival. OS was better for non-colorectal versus colorectal NENs (HR 0.34; 95% CI: 0.12-0.95) (Fig. 1B and supplementary Fig. 2).

The median PFS was 5.7 months (95% CI: 5.1–9) and the PFS rates at 6 and 12 months were 43.2% (95% CI: 29.9–62.6) and 21.6% (95% CI: 11.7–39.9), respectively (Fig. 1C). The median PFS was 5.7 months (95% CI: 4.2–not reached [NR]) and 5.7 months (95% CI: 4.9–9.3) for NETs

and NECs; and 5.4 months (95% CI: 2.7-NR) and 5.9 months (95% CI: 5.1–9.3) for Ki-67 ≤ 55 and >55%, respectively (Supplementary Fig. 2).

Multivariable cox regression analysis revealed high baseline chromogranin A (CgA) levels were associated with a lower risk of progression (HR: 0.26, 95% CI: 0.09–0.72) (Fig. 1D and supplementary Fig. 4).

Among the 37 patients evaluable for efficacy, ORR was 56.8% and disease control rate (DCR) 83.8% (Fig. 2A). Twenty-one patients (56.8%) had partial response (PR), 10 (27.0%) had stable disease (SD), and 3 (7.9%) had progressive disease (PD) as their best response. Three patients who died due to disease progression before the first tumor assessment were not evaluated for response but were considered as treatment failures and included in the denominator for all ORR and DCR calculations. Twenty-four patients (64.9%) experienced a reduction in tumor size from baseline, with sustained response for more than 12 months in five patients (13.5%) (Fig. 2B). One of the two patients with PD-L1-positive tumors and the patient with MSI tumor had sustained response. Characteristics of patients with long-term responses are detailed in Table 2. The median duration of response (DoR) was 6.4 months (range: 1.5–27.7+). ORR significantly differed by primary tumor site, with colorectal NENs having the worst ORR (ORR 16.7% vs 64.5%; p = 0.02) (Fig. 2C). ORR did not significantly differ by tumor differentiation or Ki-67 index, although ORR was numerically higher in patients with Ki-67 > 55% (ORR 64% vs 41.7%) and in NECs versus NETs

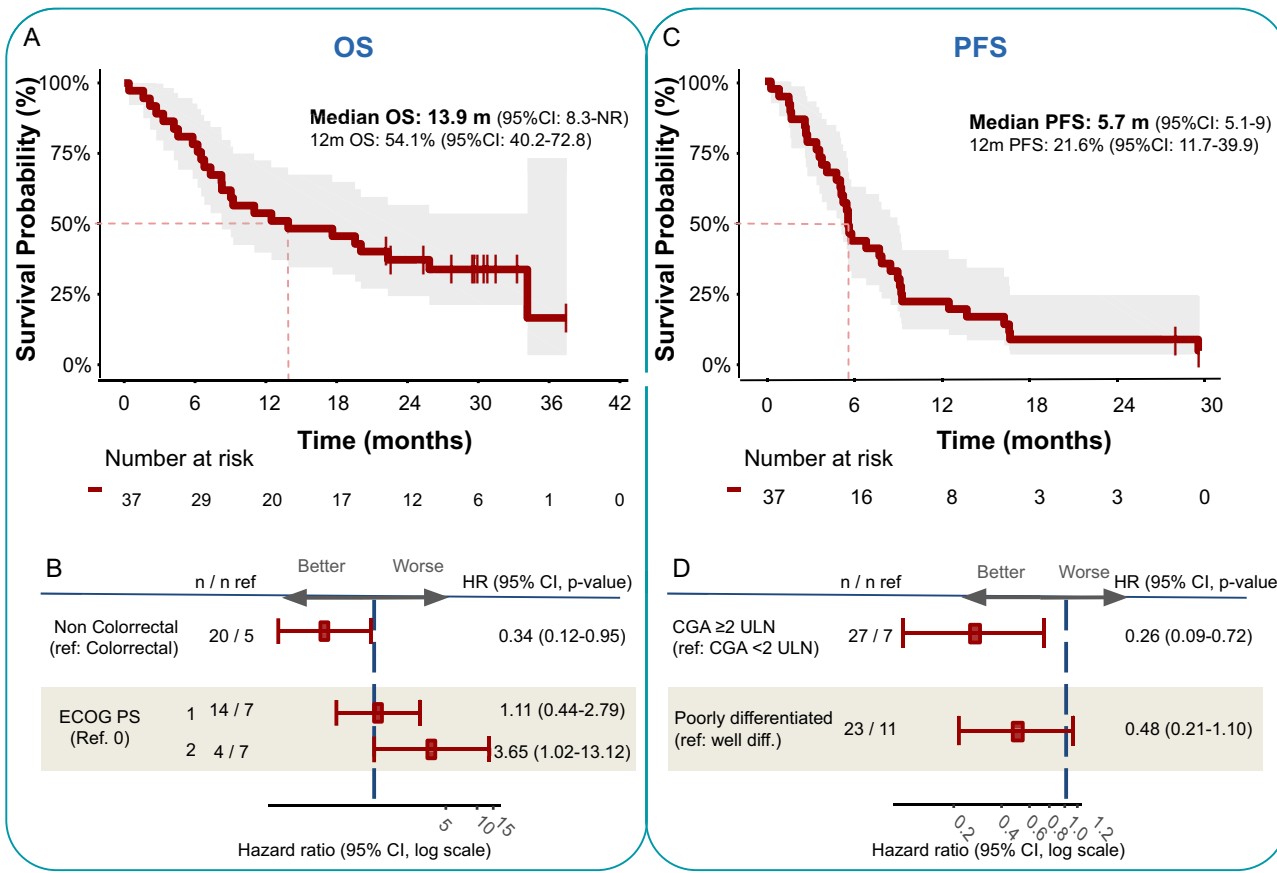

**Fig. 1 | PFS and OS of patients with high-grade NENs of GEP or UK origin treated with nivolumab, carboplatin, and etoposide. A** Kaplan Meier showing the OS for the full dataset (n = 37). The red line shows the estimated survival proportion and the shadow area the 95% CI. The red dashed lines indicate the 50% survival probability point estimate. **B** Multivariable analysis to find potential baseline prognostic factors for OS. The forest plot shows the hazard ratio of each subgroup and its 95% CI. **C** Kaplan-Meier showing the PFS for the full dataset. **D** Multivariable analysis to find potential baseline prognostic factors for PFS. The forest plot shows the hazard ratio of each subgroup and its 95% CI. Multivariable analyses were performed using the Cox model and are exploratory. Significance tests are two-sided. Source data are provided as a Source Data file. CGA chromogranin A, CI confidence interval, Diff differentiated, ECOG-PS Eastern Cooperative Oncology Group Performance Status, HR hazard ratio, OS overall survival, PFS progression-free survival.

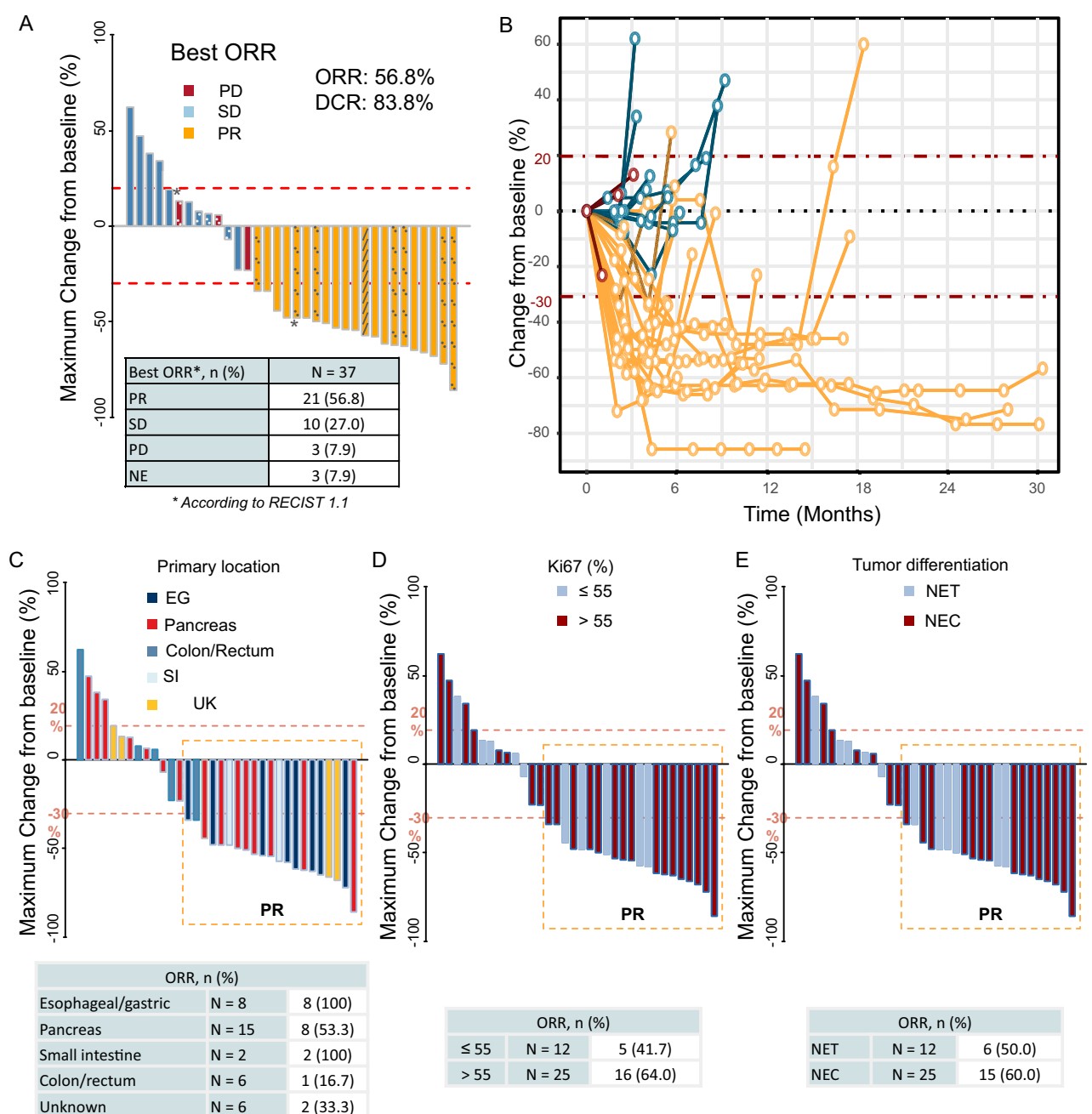

**Fig. 2 | Efficacy of Nivolumab, Carboplatin and Etoposide as first-line treatment of highgrade NENs of GEP or UK origin. A** Waterfall plot showing objective response rate (ORR) and the percentage of maximum change from baseline tumor size for each patient (*n* = 37). **B** Spider plot showing the evolution of relative tumor size from the first dose of study treatment until the last tumor evaluation (*n* = 37). Asterisk sign indicates PD-L1 positivity, lined plotted columns indicate MSI, dotted plotted columns MSS. Maximum change in tumor size, shown as percentage from baseline, and ORR rates analyzed by subgroups clustered according to baseline characteristics such as primary tumor site (**C**), Ki-67 proliferation index (**D**), or tumor differentiation (**E**). Three patients, one MSS and two unknown died due to disease progression before the first tumor assessment were not evaluated for response. Exploratory subgroup analyses were performed using Fisher's Exact Test (**C**) and Pearson's Chi-squared test (**D** and **E**). Significance tests are two-sided. Source data are provided as a Source Data file. DCR disease control rate, EG esophageal and gastric, NE not evaluable, NEC neuroendocrine carcinoma, NET neuroendocrine tumor, ORR objective response rate, PD progressive disease, PR partial response, RECIST Response Evaluation Criteria In Solid Tumors, SD stable disease, SI small intestine, UK unknown.

(60% vs 50%) (Fig. 2D, E). Patients with baseline CgA levels >2-times the upper limit of normal (2xULN) showed higher ORR rates (63% vs. 28.6%), although this difference was not statistically significant.

Subgroup analysis to explore potential signs of differential efficacy in terms of OS, PFS, and ORR by most relevant clinical features, including age, sex, performance status, tumor differentiation, Ki-67 range, plasma levels of tumor markers, tumor location, MSI status,

PDL1 expression, were preplanned although the study was not powered for formal comparisons.

**Safety**

The median duration of treatment was 3.5 months (95% CI: 3-3.7) for platinum-based chemotherapy and 4.4 months (95% CI: 3.8-7.1) for nivolumab. Twenty-three patients (57%) received 6 cycles of

**Table 2 | Baseline patient characteristics are shown for the full dataset and by the duration of response (DoR <versus ≥12 months)**

| Characteristics | | G3 NENs N = 37 | DoR < 12 m N = 32 | DoR ≥ 12 m N = 5 |
|---|---|---|---|---|
| Age, years | Median (range) | 61 (28–84) | 61 (28–84) | 60 (44–80) |
| Gender, n (%) | Male | 25 (67.6) | 22 (68.8) | 3 (60.0) |
| | Female | 12 (32.4) | 10 (31.2) | 2 (40.0) |
| ECOG PS, n (%) | 0 | 11 (29.7) | 10 (31.2) | 1 (20.0) |
| | 1 | 22 (59.5) | 18 (56.2) | 4 (80.0) |
| | 2 | 4 (10.8) | 4 (12.5) | 0 (0.0) |
| Stage at diagnosis, n (%) | I | 1 (2.7) | 1 (3.1) | 0 (0.0) |
| | III | 1 (2.7) | 0 (0.0) | 1 (20.0) |
| | IV | 35 (94.6) | 31 (96.9) | 4 (80.0) |
| Differentiation, n (%) | NET | 12 (32.4) | 11 (34.4) | 1 (20.0) |
| | NEC | 25 (67.6) | 21 (65.6) | 4 (80.0) |
| Ki 67, n (%) | 21 – 55% | 12 (32.4) | 11 (34.4) | 1 (20.0) |
| | >55% | 25 (67.6) | 21 (65.6) | 4 (80.0) |
| Primary site, n (%) | Esophageal | 2 (5.4) | 2 (6.2) | 0 (0.0) |
| | Gastric | 6 (16.2) | 4 (12.5) | 2 (40.0) |
| | Pancreatic | 15 (40.5) | 13 (40.6) | 2 (40.0) |
| | Colonic | 4 (10.8) | 4 (12.5) | 0 (0.0) |
| | Rectal | 2 (5.4) | 2 (6.2) | 0 (0.0) |
| | Small intestine | 2 (5.3) | 1 (3.1) | 1 (20.0) |
| | Unknown | 61 (16.2) | 6 (18.8) | 0 (0.0) |
| Metastatic sites number, n (%) | 1 | 10 (27.0) | 8 (25.0) | 2 (40.0) |
| | ≥2 | 27 (73.0) | 24 (75.0) | 3 (60.0) |
| Metastasis sites, n (%) | Liver | 31 (83.8) | 29 (90.6) | 2 (40.0) |
| | Lung | 9 (24.3) | 9 (28.1) | 0 (0.0) |
| | Lymph nodes | 18 (48.6) | 15 (46.9) | 3 (60.0) |
| | Bone | 10 (27.0) | 10 (31.2) | 0 (0.0) |
| Previous surgery, n (%) | Yes | 6 (16.2) | 4 (12.5) | 2 (40.0) |
| | No | 30 (81.1) | 27 (84.4) | 3 (60.0) |
| | Unknown | 1 (2.7) | 1 (3.1) | 0 (0.0) |
| CgA, n (%) | <2x ULN | 7 (18.9) | 7 (21.9) | 0 (0.0) |
| | ≥2x ULN | 27 (73.0) | 24 (75.0) | 3 (60.0) |
| | Unknown | 3 (8.1) | 1 (3.1) | 2 (40.0) |
| Enolase, n (%) | <2x ULN | 13 (35.1) | 11 (34.3) | 2 (40.0) |
| | ≥2x ULN | 21 (56.8) | 19 (59.4) | 2 (40.0) |
| | Unknown | 3 (8.1) | 2 (6.2) | 1 (20.0) |
| LDH, n (%) | <2x ULN | 9 (24.3) | 8 (25.0) | 1 (20.0) |
| | ≥2x ULN | 28 (75.7) | 24 (75.0) | 4 (80.0) |
| PD-L1; n (%) | Positive | 2 (5.4) | 2 (6.2) | 0 (0.0) |
| | Negative | 31 (83.8) | 26 (81.3) | 5 (100.0) |
| | Unknown | 4 (10.8) | 4 (12.5) | 0 (0.0) |
| MSI; n (%) | Positive | 1 (2.7) | 0 (0.0) | 1 (20.0) |
| | Negative | 13 (35.1) | 10 (31.3) | 3 (60.0) |
| | Unknown | 23 (34.3) | 22 (68.8) | 1 (20.0) |

chemotherapy as scheduled. Two patients (5.4%) completed 24 months of nivolumab maintenance therapy.

Grade ≥3 toxicities occurred in 23 (60.5%) patients (Fig. 3, Supplementary Tables 1 and 2). The most frequent grade ≥ 3 treatment-related adverse events were neutropenia (39.5%), febrile neutropenia (10.5%), anemia (7.9%), fatigue (7.9%) and thrombopenia (5.2%) (Fig. 3A). Most toxicities emerged and were more severe during the induction phase (Fig. 3B). The majority of immune-mediated adverse events were G1-2 (Supplementary Table 1). One patient (2.6%)

experienced grade 3 ALT increase and one patient had a grade 3 acute kidney injury considered immune-related; both resolved without sequelae. There was one treatment-related death, which occurred after the first cycle of therapy in the induction phase. The event was a grade 4 pancytopenia and esophageal mucositis, bacteremia due to *Escherichia Coli*, septic shock and severe upper gastrointestinal bleeding that led to a fatal outcome.

## Discussion

This phase II study evaluated the combination of an ICI and standard first-line chemotherapy in high-grade GEP NENs. Nivolumab in combination with carboplatin and etoposide achieved an ORR of 56.8%, a DCR of 83.8%, and a median survival of 13.9 months[3,19–22]. Responses were profound and durable, with about one-third of responses maintained beyond 12 months and a 2-year OS rate of 37.6%. Although the primary endpoint of this trial was not met (1-year OS rate of 72%), the high proportion of long-term survivors is in our opinion particularly encouraging considering that our cohort was mostly composed of patients with multiple adverse prognostic features, such as poor histological differentiation (68%), Ki-67 index > 55% (66%), stage IV at diagnosis (95%), ≥ 2 metastatic sites (74%), poor performance status (10.8% ECOG 2) and high baseline enolase levels (63%). In addition, our cohort had a low prevalence of molecular alterations predictive of response to immunotherapy, such as positive PD-L1 or MSI, and the majority presented liver involvement at study entry, which can co-opt immune tolerance mechanisms to induce immunotherapy resistance[23].

The standard of care for patients with advanced or metastatic G3 GEP NENs is systemic chemotherapy. Cisplatin or carboplatin in combination with etoposide are the most widely used regimens, that yield ORR of 31%, DCR of 64%, and a median survival of 11 months, as reported in one of the benchmark studies, the NORDIC NEC[19]. Response to chemotherapy was lower in G3-low GEP NENs (ORR 15% versus 42% in NENs with Ki67 21–55% versus ≥ 55%), although they had better survival compared to G3-high GEP NENs (14 versus 10 months, respectively). More recently, two randomized Asian trials that compared the combination of cisplatin and etoposide with cisplatin and irinotecan, showed similar efficacy for both regimens, with somewhat higher ORR (53-63%) than those reported in the NORDIC NEC study, but no improvement in OS (10.2–12.5 months)[20,21] and a low rate of long-term survivors (2-year OS rate ≤ 20%). Another recent randomized trial in a Western population (ECOG ACRIN EA2142) compared platinum and etoposide versus CAPTEM in non-small cell G3 GEP-NENs (including NETs and NECs). This trial was closed for futility, concluding that CAPTEM was not associated with increased efficacy. It reported an ORR of 22%, a median PFS of 5.4 months, and a median OS of 10.6 months for the carboplatin and etoposide combination[22]. In this context, the results of our study combining ICI with first-line platinum-based chemotherapy are encouraging. However, it should be noted that different proportions of relevant prognostic features in study populations, particularly regarding tumor differentiation, Ki-67 index or primary tumor site, limit comparisons with benchmark studies.

ICI has been incorporated as the standard of care for other high-grade NENs (SCLC, MCC), although its use in G3 digestive NENs is still a matter of debate. Early trials conducted in heavily pretreated populations showed overall disappointing results. However, they suggest a potential role in certain molecularly selected subgroups, such as MSI/dMMR, ARID1A-mutated, or TMB-H tumors[8,9,24]. Dual PD1/CTLA4 blockade showed promising preliminary activity in small cohorts of patients with high-grade NENs enrolled in two basket trials (DART SWOG 1609 and CA209-538)[25,26] and is in fact one of the recommended treatment options in the NCCN guidelines for extrapulmonary NECs progressed to first-line chemotherapy[27]. However, clinical trials specifically conducted in GEP NENs have shown more modest results. For instance, the DUNE trial that assessed durvalumab and tremelimumab in a cohort of G3 GEP NENs progressed to platinum-based first-line

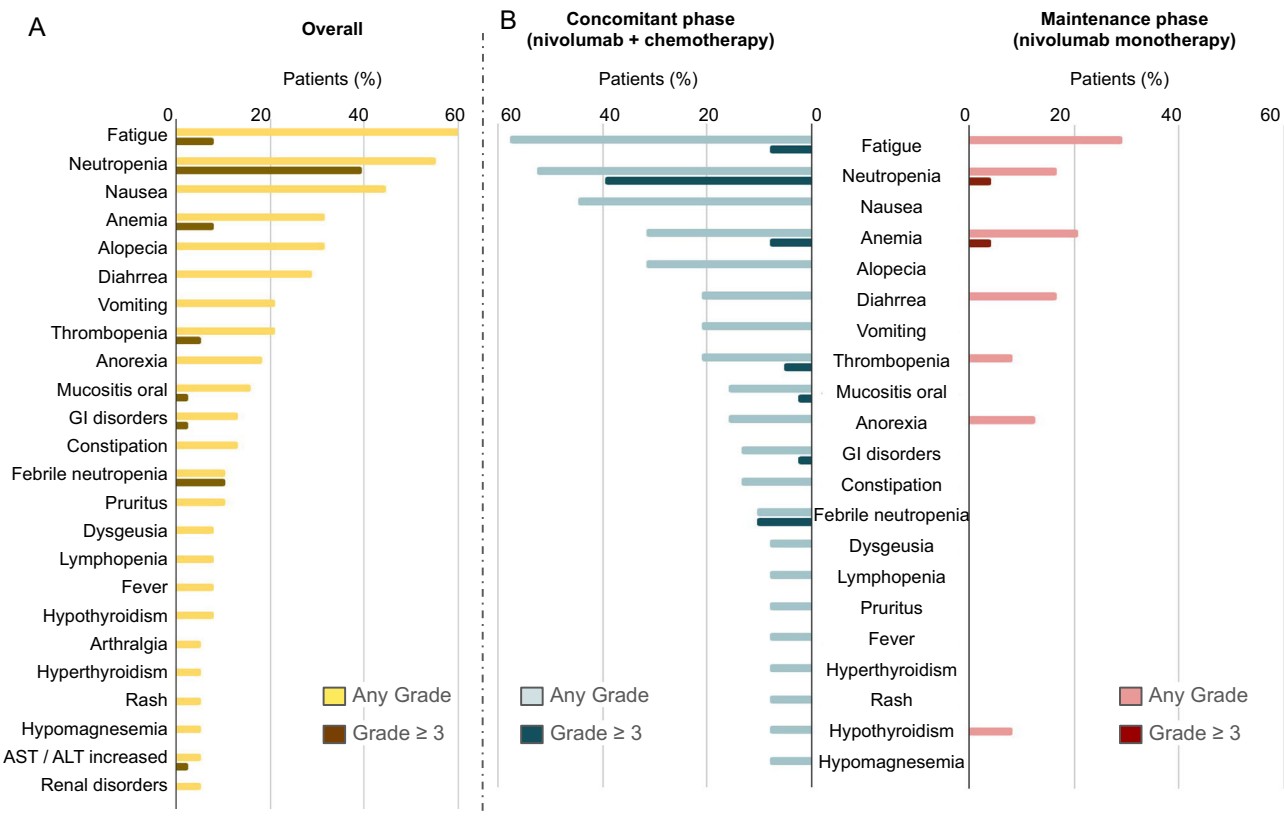

**Fig. 3 | Safety profile of nivolumab in combination with carboplatin and etoposide. A** Treatment-related adverse events encountered in >5% of patients. The graph represents the worst grade per patient. The percentage of patients experiencing an event is depicted (*n* = 37). **B** Distribution of toxicities by study phase (induction versus maintenance phase) from first dose of study treatment until last follow-up (*n* = 37 for concomitant phase and *n* = 24 for maintenance phase). Source data are provided as a Source Data file.

therapy, reported an ORR of 9.1% and a 9-month OS rate of 36.1%[28]. Consistent with our study, outcomes did not substantially differ by tumor differentiation (NET vs. NEC) or Ki-67 index, and about one-third of patients were long-term survivors. In fact, the G3 GEP NEN cohort of the DUNE trial surpassed the pre-established OS futility threshold, which was the primary endpoint. Similar results were reported in the larger NIPINEC study (GCO-001), a randomized non-comparative trial that allocated patients with NECs, including 93 with advanced large-cell lung cancer and 92 with GEP, all having progressed to 1 or 2 prior lines of therapy including platinum-based chemotherapy, to receive nivolumab or nivolumab plus ipilimumab[15]. The combination of nivolumab and ipilimumab reached its primary endpoint of ORR at 8 weeks >10%. Dual ICI induced higher ORR in lung NECs (18.2%) than in GEP NECs (11.6%). Median PFS (1.9 months) and OS (5.8 months) were not encouraging, although further follow-up and molecular profiling are needed to identify whether a subset of patients may achieve long-term benefit.

Chemotherapy may improve the immunological effects of ICIs as it induces an immunogenic cell death, reduces regulatory T-cell activity, and induces PD-L1 expression, and this synergy has in fact been demonstrated in the clinical setting[6,7]. Moreover, chemotherapy has short-term cytotoxic effects that may slow down tumor growth and allow enough time for ICI to induce/re-invigorate an effective antitumor immune response. Results are particularly encouraging in G3 GEP NENs of non-colorectal origin (ORR 70.8%), although small numbers and lack of control preclude definitive conclusions. Conversely, colorectal NENs showed limited benefit from the combination of ICI with chemotherapy, with particularly poor outcomes in line with those reported by Sorbye et al. for first-line platinum/etoposide treatment in this subgroup of patients[29]. Other features associated with increased ORR were increased baseline CgA levels and poor histological differentiation, although these correlations did not reach statistical significance. CgA levels have been classically associated with poor prognosis in GEP-NETs, mostly as they are related to tumor bulk[30,31]. Nevertheless, in the context of G3 NENs its prognostic role is more complex, as the lack of CgA expression may be associated with poorly differentiated, more aggressive tumors with a worse outcome. Consistently, CgA was associated in our study with significantly improved PFS (median of 5.9 and 3.9 months for CgA ≥2 ULN vs <2 ULN, respectively). More interestingly, CgA may play an immunomodulatory role as it is capable of boosting the innate immune system by inducing TNFα secretion and is correlated with immune cell infiltration in chronic inflammatory conditions such as lymphocytic colitis[32,33]. The potential interaction of CgA with the regulation of immune response deserves to be further explored. Regarding tumor differentiation, our observations and evidence from other trials suggest that NECs obtain greater benefit from ICIs than NETs, possibly due to the higher mutational burden and enhanced neoantigen presentation of poorly differentiated NECs. In fact, 4 out of 5 long-term responders were NECs. However, a low proportion of PD-L1 expression was observed in our cohort (2 patients, 5.4%) and all patients with prolonged duration of response were PD-L1 negative. Similar observations were reported in the DUNE trail, where no correlation was found between PD-L1 expression and efficacy[28].

The addition of nivolumab to standard chemotherapy did not reveal any safety concerns. Most frequent grade ≥3 toxicities primarily occurred during the induction phase and were mainly associated to platinum-based chemotherapy (neutropenia (39.5%), febrile neutropenia (10.5%), anemia (7.9%), fatigue (7.9%) and thrombopenia (5.2%)). Immune-mediated adverse events were generally mild and

manageable. Treatment discontinuation rates were similar to those reported in previous studies, and primarily resulted from disease progression.

The main limitation of this trial is the fact that it was a non-randomized study. The lack of a parallel control group restricts the ability to directly compare the treatment outcomes with a standard arm. In addition, the study sample size limits the statistical power to conduct exploratory clinical or molecular analysis to identify subgroups of patients that may obtain a greater benefit, particularly relevant due to the heterogeneity of the study population in terms of primary tumor site and tumor differentiation. These limitations highlight the need for larger, controlled studies to validate our findings and further investigate the added value of combining an ICI with standard chemotherapy. Addressing these limitations, the S2101 (NCT05058651) phase II/III trial, currently underway, aims to enroll 189 patients with extrapulmonary NECs. It will specifically evaluate the impact of adding atezolizumab to standard chemotherapy (carboplatin/cisplatin plus etoposide) compared to chemotherapy alone. This study will help to determine the potential added value of combining ICIs with standard first-line chemotherapy in advanced extrapulmonary G3 NENs.

In conclusion, this study assessed the addition of ICIs to platinum-based chemotherapy in chemonaïve patients with advanced or metastatic G3 NENs of GEP or UK origin. Although the primary endpoint (12-month OS rate) was not met, the high proportion of long-term survivors (37.6% OS rate at 2 years) is encouraging in the context of a highly aggressive disease. These findings justify further exploration in larger, randomized trials to more accurately assess the risk-benefit balance of combining immunotherapy with platinum-based chemotherapy as a first-line treatment for patients with high-grade GEP-NENs. Translational studies will also be crucial to identify predictive biomarkers to improve the selection of patients who may benefit the most from this therapeutic strategy.

## Methods
### Study design and participants
The NICE-NEC trial was a multicenter, single-arm, phase II clinical trial that recruited patients across 12 centers belonging to the Spanish Taskforce of Neuroendocrine Tumors (GETNE). Key inclusion criteria included histologically confirmed, unresectable advanced or metastatic NENs of GEP or unknown primary origin, G3 (Ki-67 > 20% or mitotic rate >20 per 10 high-power fields (HPF)), chemotherapy-naive status, measurable disease, adequate organ function, and an ECOG performance status of 0-2. Neither sex nor gender were considered to design the study but sex was registered as a relevant covariable for univariable and multivariable analysis. Sex was registered by investigators based on biological grounds. Gender was not registered. Available fresh or archived formalin-fixed, paraffin-embedded tumor tissue was required in those patients that provided additional (non-mandatory) consent for translational studies.

Paraganglioma, adrenal, thyroid, parathyroid, or pituitary endocrine tumors, and large or small cell NEC of the lung were excluded. Patients who had previously received ICI, had undergone organ transplantation, were on systemic chronic steroid therapy (>10 mg/day prednisone or equivalent) or taking other immunosuppressive agents, or had received any investigational drug within 28 days prior to the start of the study treatment were also excluded. Additionally, patients with a known history or active/chronic infection with hepatitis B or C virus or human immunodeficiency virus (HIV), or with other significant infections requiring medication were not eligible for the trial.

The study protocol was approved by the competent authority in Spain and the Institutional Review Board of Hospital Universitario 12 de Octubre; ref number: 19/291. The first patient was enrolled on November 4th, 2019 and the last patient on January 26th, 2021. This study was conducted in accordance with the Declaration of Helsinki

and the International Conference on Harmonization guidelines for Good Clinical Practice. All participants provided written informed consent prior to study entry. Patients were not compensated for participating in the study. This study was registered at EudraCT (2019-001546-18) on 2019-Aug-19 and www.clinicaltrials.gov (NCT03980925; https://classic.clinicaltrials.gov/ct2/show/NCT03980925) on 2019-Jun-07.

### Treatment schedule
The investigational treatment consisted of an initial induction phase of 6 cycles of nivolumab and chemotherapy, followed by maintenance treatment with nivolumab for up to 2 years in the absence of disease progression, death, unacceptable toxicity or consent withdrawal. During the induction phase, nivolumab 360 mg was administered intravenously (iv) on day 1, carboplatin AUC = 5 iv on day 1 and etoposide 100 mg/m²/day iv on days 1-3, every 3 weeks. In the maintenance phase nivolumab 480 mg was administered iv on day 1, every 4 weeks for up to 2 years.

### Study endpoints and procedures
Clinical assessments were performed every 3 weeks during the induction phase and every 4 weeks thereafter until disease progression, death, or end of study due to unacceptable toxicity or consent withdrawal. These included the review of medical history, directed anamnesis (including adverse events assessment), physical examination (including ECOG PS, vital signs, height, and weight), laboratory tests (including tumor markers CgA and enolase), concomitant medication and treatment compliance. Tumor imaging assessments were performed by computed tomography (CT) or magnetic resonance imaging (MRI) scans at baseline, every 8 weeks during the first year, and every 12 weeks thereafter until disease progression was confirmed, or the patients initiated an alternative treatment. CT, or MRI scans were assessed locally by investigators following both, RECIST 1.1, and irRECIST 1.1 criteria.

The primary objective of this study was the 12-month OS rate. OS was determined from the time of the initiation of the study treatment to the date of death from any cause.

Secondary endpoints included ORR, DCR, DoR, PFS, and safety. ORR was defined as the percentage of patients who achieved a complete (CR) or PR throughout the study period. DCR was defined as the percentage of patients achieving CR, PR, or SD. DoR was defined as the time elapsed from the date of the first documentation of an objective response (CR or PR) to the date of progressive disease. PFS was defined as the time elapsed from the first dose of the study treatment until the date of progressive disease or death, whichever occurred first. Safety was based on the assessment of AEs, clinical laboratory test results, vital signs, and physical examinations. AEs and laboratory values were graded according to the Common Terminology Criteria for Adverse Events v.5.0 (CTCAE). Pre-planned subgroup analyses were performed to explore potential signs of differential efficacy by most relevant clinical features, although the study was not powered for formal comparisons; therefore, all findings are exploratory in nature.

### PDL1 expression assessment
To assess the expression of PDL1, IHC was performed on whole formalin-fixed paraffin-embedded tissue specimens (FFPE) freshly cut tissue sections of 4-µm thickness using an automated stainer (Histocore Spectra ST Leyca Biosystems). The primary PDL1 antibody used was clone 22C3 (Agilent/Dako, PharmDx; Supplementary Table 3). For optimisation of the PD-L1 staining, samples from PD-L1-positive squamous cell lung carcinoma (SCC) were included as external positive control.

Thirty-five tumor samples that had at least 50-100 viable cells were considered adequate for PDL1 expression assessment. Immunostains were evaluated by pathologists blinded to the clinical data.

PDL1 combined positive score (CPS) was calculated as the proportion of PDL1-staining cells (tumor cells (TCs) and tumor-infiltrating immune cells (ICs) with positive membranous staining of any intensity) relative to all viable tumor cells. PDL1 tumor proportion score (TPS) was calculated as the percentage of viable tumor cells showing partial or complete membrane staining at any intensity. A TPS/CPS < 1% was considered negative.

## Characterization of mismatch-repair (MMR)/microsatellite instability (MSI)

MMR/MSI was determined in patients with available extra tumor tissue ($N = 14$). Two techniques were used to determine the MMR/MSI status: Immunohistochemistry (IHC) and polymerase chain reaction (PCR). IHC staining for MLH1, PMS2, MSH2, and MSH6 was performed on formalin-fixed, paraffin-embedded (FFPE) tumor tissue blocks by standard automated staining methods performed in each Hospital (Supplementary Table 3). MMR IHC results were interpreted and reported by clinical pathologists. For the assessment of MSI status by PCR-based fragment-sizing test, five microsatellite mononucleotide loci, namely BAT25, BAT26, NR21, NR24, and MONO27, were analyzed using OncoMate MSI Dx Analysis System according to the manufacturer's protocol (REF: MD3140; https://www.promega.es/products/microsatellite-instability-msi-testing/clinical-msi-testing-ivd/oncomate-msi-dx-analysis-system/; Promega Corporation, Madrid, Spain). Briefly, the 5 microsatellite markers were amplified using a multiplex fluorescence PCR and subjected to capillary electrophoresis on SeqStudio Genetic Analyzer (Thermo Fisher Scientific, Madrid, Spain). MSI was scored when at least 2 of 5 loci were unstable.

## Statistical analysis

Sample size was calculated using a two-sided one arm survival test (https://stattools.crab.org/Calculators/oneNonParametricSurvival.htm)[34]. The study aimed for a statistical power of 80% and an alpha error of 0.05. The null hypothesis was based on historical controls and assumed a 12-month OS rate of 50%, while the alternative hypothesis proposed an OS rate of 72%. The accrual time for patient enrollment was 18 months, followed by a 12-month follow-up period. Considering a dropout rate of 10%, a final sample size of 38 patients was determined for the study.

Efficacy analysis was based on the full analysis set that included all enrolled eligible patients. The safety analysis set comprised all patients who received at least one dose of the study treatment including those who were ineligible or experienced disease progression or death before the first on-treatment scan.

Continuous variables were summarized using descriptive statistics (n, mean, standard deviation, range, and median). Frequency counts, and the percentage of subjects within each category were provided for categorical data. Response rates were estimated using 95% confidence intervals (CI), or range (minimum to maximum values) intervals. Survival or time-to-event endpoints were estimated using the Kaplan–Meier method, and Cox regression analysis to obtain hazard ratios and CIs. Patients without documented progression or death by the time of the analysis were censored at the last date of tumor evaluation for PFS assessment. For OS assessment, patients without documented death were censored at the last date of follow-up. All statistical tests were considered two-tailed, and results with $p < 0.05$ were considered significant. Variables assessed in univariable subgroup analysis to explore potential signs of differential efficacy by most relevant clinical features included the following: primary tumor site, tumor differentiation, Ki-67 index, ECOG performance status, sex, age, chromogranin A, neurospecific enolase, and LDH. Multivariable regression models also assessed these variables to analyze their potential relationship with efficacy endpoints. To obtain the multivariable models, we employed the 'backward' stepwise selection method, systematically eliminating non-significant exploratory variables until a significant set was achieved. In instances where a significant set could not be reached, the last two remaining variables were included for analysis (see supplementary information for further details). All statistical analyses were performed with R and SPSS (IBM SPSS Statistics Version 26, Armonk, NY). Figures and tables were generated using RStudio (Version 1.2.5033 2009-2019 RStudio, Inc., Boston, MA, US).

## Reporting summary

Further information on research design is available in the Nature Portfolio Reporting Summary linked to this article.

## Data availability

The study protocol is available as Supplementary Note in the Supplementary Information file. The clinical raw data are protected and are not available due to data privacy laws. The data that support the findings of this study are available from the corresponding author upon reasonable request (equivalent purposes to those for which the patients grant their consent to use the data). Data sharing requests will be considered on a case-by-case basis in a timely manner. Response to access requests will be provided within 4 weeks and data will be available for 6 months once access has been granted. Data will be provided anonymized, with no personal identifiable data. A Source data has been provided with all relevant raw data from each figure or table of the main manuscript and Supplementary Information. The remaining data are available within the Article, Supplementary Information or Source Data file. Source data are provided with this paper.

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

## Acknowledgements

This work was sponsored by the Grupo Español de Tumores Neuroendocrinos y Endocrinos (GETNE). Bristol-Myers Squibb (BMS) provided nivolumab and awarded a grant to GETNE to pay the costs of the study. The funder did not have a role in designing or conducting the study, and was not involved in the analysis and interpretation of study results. Dr. Riesco-Martinez would like to acknowledge the 19th edition of the Methods in Clinical Cancer Research Workshop, organised by ESMO-EORTC-AACR and held in Zeist, Netherlands in 2017, in which the protocol of this trial was developed, for the excellent academic and scientific support provided. The authors thank all patients and families, investigators, and study staff involved in the NICE-NEC trial; the MFAR Clinical Research team for regulatory, monitoring, and quality assurance activities; Pau Doñate Ph.D. for manuscript and language editing; and Emilio Pecharromab MsC. for statistical support.

## Author contributions

R.G.C. and M.C.R.M. were responsible for the study design and coordination. All authors (R.G.C., M.C.R.M., J.C., V.A., P.J.-F., A.T., E.G., I.S., M.B., T.A.-G., A.C., B.A.-P., J.H., E.P., O.A.C.-T., A.L.-P., A.Tei, Y.R.-G. and B.S.) made substantial contributions to data acquisition, interpretation of study results, manuscript drafting or reviewing it critically for important intellectual content, provided approval to the final version to be published and agreed to be accountable for all aspects of the work and ensure that questions related to the accuracy or integrity of any part of the work were appropriately investigated and resolved.

## Competing interests

RGC has received honoraria for speaker engagements, advisory roles or funding for continuous medical education from AAA, Advanz Pharma, Amgen, Astellas, Bayer, BMS, Boehringer, Esteve, GSK, Hutchmed, Ipsen, Midatech Pharma, MSD, Novartis, PharmaMar, Servier, Takeda, and has received research support from Pfizer, BMS and MSD. EG has received honoraria for speaker engagements, advisory roles or funding of continuous medical education from Adacap, AMGEN, Angelini, Astellas, Astra Zeneca, Bayer, Blueprint, Bristol Myers Squibb, Caris Life Sciences, Celgene, Clovis-Oncology, Eisai, Eusa Pharma, Genetracer, Guardant Health, HRA-Pharma, IPSEN, ITM-Radiopharma, Janssen, Lexicon, Lilly, Merck KGaA, MSD, Nanostring Technologies, Natera, Novartis, ONCODNA (Biosequence), Palex, Pharmamar, Pierre Fabre, Pfizer, Roche, Sanofi-Genzyme, Servier, Taiho, and Thermo Fisher Scientific. EG has received research grants from Pfizer, Astra Zeneca, Astellas, and Lexicon Pharmaceuticals. TA-G declares participating in advisory boards for Astellas, Bayer, Bristol Myers Squibb, EISAI, IPSEN, Lilly, Novartis Advanced Accelerator Applications, Pfizer, Roche, and Sanofi; act as invited speaker for Janssen-Cilag; and being project lead for Johnson & Johnson, IPSEN and Pfizer. JC declares scientific consultancy role

(speaker and advisory roles) for Novartis, Pfizer, Ipsen, Exelixis, Bayer, Eisai, Advanced Accelerator Applications, Amgen, Sanofi, Lilly, Hudchmed, ITM, Merck Serono, Roche, Esteve, Advanz; and received research grants from Novartis, Pfizer, Astrazeneca, Advanced Accelerator Applications, Eisai, Amgen, ITM and Bayer. PJ-F has provided scientific advice and/or received honoraria from Astellas, BMS, Eisai, Lilly, MSD, Pfizer and Rovi. IS received payment or honoraria for lectures, presentations, speakers bureaus, manuscript writing or educational events from AAA, Novartis, Pharmamar, Ipsen, Pfizer, Amgen,Bayer; support for attending meetings and/or travel from AAA, Novartis, Pharmamar, Ipsen, Boehringer, Advanz Pharma, Esteve Pharmaceuticals, Amgen, Bayer; and participated on a Data Safety Monitoring Board or Advisory Board for AAA, Novartis, Pharmamar, Ipsen, Boehringer, Advanz Pharma, Esteve Pharmaceuticals, Amgen, Bayer. AT received support for attending meetings and /or travel from MSD, Roche; received payment honoraria for lectures/scientific advice from Pfizer and Diaceutics; and working as an employee at MD Anderson Hospital. All the remaining co-authors state that they do not have conflict of interest.

## Additional information

Maria Carmen Riesco-Martinez[1], Jaume Capdevila[2], Vicente Alonso[3], Paula Jimenez-Fonseca[4], Alex Teule[5], Enrique Grande[6,7], Isabel Sevilla[8], Marta Benavent[9], Teresa Alonso-Gordoa[10], Ana Custodio[11], Beatriz Anton-Pascual[1], Jorge Hernando[2], Eduardo Polo[3], Oscar Alfredo Castillo-Trujillo[4], Arantza Lamas-Paz[12], Ana Teijo[13], Yolanda Rodriguez-Gil[13], Beatriz Soldevilla[12] & Rocio Garcia-Carbonero[1,12,14] ✉

[1]Medical Oncology Department. Hospital Universitario 12 de Octubre, Madrid, Spain. [2]Medical Oncology Department. Vall Hebron University Hospital, Vall Hebron Institute of Oncology (VHIO), Barcelona, Spain. [3]Medical Oncology Department. Instituto Aragonés de Investigación Sanitaria, Hospital Universitario Miguel Servet, Zaragoza, Spain. [4]Medical Oncology Department. Hospital Universitario Central de Asturias, ISPA, Oviedo, Spain. [5]Medical Oncology Department., Institut Català d'Oncologia (ICO) - IDIBELL, L'Hospitalet del Llobregat, Spain. [6]Medical Oncology Department, MD Anderson Cancer Center Madrid, Madrid, Spain. [7]Facultad de Medicina, Universidad Francisco de Vitoria, Madrid, Spain. [8]Medical Oncology Department, Investigación Clínica y Traslacional en Cáncer / Instituto de Investigaciones Biomédicas de Málaga (IBIMA) / Hospitales Universitarios Regional y Virgen de la Victoria de Málaga, Málaga, Spain. [9]Medical Oncology Department, University Hospital Virgen del Rocío, Instituto de Biomedicina de Sevilla (IBIS), Seville, Spain. [10]Medical Oncology, Hospital Universitario Ramón y Cajal, Madrid, Spain. [11]Medical Oncology Department, Hospital Universitario La Paz, Madrid, Spain. [12]Center of Experimental Oncology. Gastrointestinal and Neuroendocrine Tumors Research Group. Hospital 12 de Octubre Research Institute (imas12), Madrid, Spain. [13]Pathology Department, Hospital Universitario 12 de Octubre, Madrid, Spain. [14]Facultad de Medicina, Departamento de Medicina, Universidad Complutense de Madrid (UCM), Madrid, Spain. ✉ e-mail: rgcarbonero@gmail.com

