## [Peer Review File · Nature Communications]

Nivolumab plus platinum-doublet chemotherapy in treatment-naive patients with advanced grade 3 Neuroendocrine Neoplasms of gastroenteropancreatic or unknown origin: the multicenter phase 2 NICE-NEC trial (GETNE-T1913)REVIEWER COMMENTS

Reviewer #1 (Remarks to the Author): with expertise in neuroendocrine tumors, therapy

The authors are to be commended for this first report of combination chemoimmunotherapy in patients with advanced G3 NENs, and this manuscript is yet another testament to the capabilities of the GETNE group and their exceptional spirit of collaboration to address critical questions in our field. The data from this study are absolutely essential for the neuroendocrine field, and should be disseminated through this publication.

There are a few challenges with this study design and results that ought to be discussed in the spirit of maximum transparency, and I would encourage the authors to revise accordingly:

1. The primary endpoint was 12-month landmark PFS, and the null hypothesis was not excluded. The authors' conclusion that further study is necessary is reasonable, but explicitly offering this context is appropriate.
2. Given the established relationship between Ki-67 and survival, comparisons between populations of patients with different proportions of patients with NETs (with generally lower Ki-67s) is fraught. Explicit accounting for this in both study design and results would be helpful to readers so they can contextualize and interpret the results.
3. Accordingly, Supplementary Table 1 may merit inclusion in the primary manuscript, rather than relegation to the supplement. Most of the durable responses were in patients with PD-NEC that was TMB low and PD-L1 negative. This is exactly the population of patients one would hope to benefit with this combination approach, and I would submit that this may be the most exciting observation of the manuscript.

Overall, the work presented in this manuscript is timely and important. I believe that with some additional detail to support transparent interpretation, it is appropriate for publication.

Daniel M. Halperin, MD

Reviewer #2 (Remarks to the Author): with expertise in biostatistics, clinical trial study design

General comments:

1. The abstract should clearly state whether the primary endpoint was met. Since the primary endpoint was not met, I don't understand why authors stated "Randomized trials are required to confirm these promising results."
2. The abstract should also include trial design parameters, such as type I error, power, null hypothesis and alternative hypothesis.
3. The manuscript shows lack of attention to details and care.
 - 3.1. The reporting of Table 1 and the manuscript text are inconsistent in multiple places (see specific comment).
 - 3.2. Figure 1c does not appear to be referenced anywhere. Maybe Line 149 should be 1c.
 - 3.3. Line 178, Figure 3A clearly show grade 3+ Neutropenia with more than 50%.
4. Given that this is a clinical trial report, I would prefer authors to report the primary endpoint (OS) first before all other secondary endpoint.
5. The denominator of ORR and DCR percentage calculation should be all patients who started treatment. Removing patients who died prior to disease assessment (Line 138) from the denominator will make the result appear to be better than it is. The ORR and DCR calculation as currently presented is not appropriate.
6. A lot of subgroup analysis were presented, e.g. ORR by primary location, ORR by Ki67, and ORR by differentiation. It should be clearly noted in the manuscript that all these are exploratory analysis. These are not powered; therefore, the p-value should be removed. The confidence interval of the point estimate can be presented. Same comments for PFS/OS by subgroups.
7. The multivariable Cox model may be over saturated. Given 35 PFS events, I would not incorporate 9 variables (10 degree of freedom) in the model. Same comment for OS. A reduced number of variables should be used.
8. I find it curious that the median PFS is 5.7 month but the treatment duration is only 3.5 months for platinum-based chemotherapy and 4.4 months for nivolumab while majority of patients were off-treatment due to disease progression. Can authors explain why the treatment duration is much shorter than PFS while majority of patients were off-therapy due to progression?

9. Can author provide a citation of the “two-sided one arm survival test”? There are many such tests available.

10. The decision boundary of the primary endpoint should be stated so the readers can judge whether 54% is barely miss the mark.

Specific comments:

1. Line 122, the primary site of Inguinal also has 2 which is the same as esophageal , rectal, and small intestine but Inguinal was omitted from the manuscript text. Is there a reason why?

2. Line 122, table 1 only shows 1 unknown primary but the manuscript text states 5. Please reconcile.

3. Line 126, table 1 shows 27 (73.0%) elevated GgA but the text states 79.4%. Please reconcile.

4. Line 126, enolase reporting is inconsistent between table 1 and the text.

5. Line 126, PD-L1 reporting is inconsistent between table 1 and the text.

6. Line 156, the word “multivariable” should be used, rather than “multivariate” . Same comments to line 375. Please review the manuscript as a whole to correct all instances.

7. Line 170, please remove “trend toward” the point estimate is either statistical significant or not.

8. Line 374, I don’t understand what it means by “full range intervals”.

Reviewer #3 (Remarks to the Author): with expertise in neuroendocrine tumors, therapy

The authors here reported an interesting phase II study on grade 3 (G3) neuroendocrine neoplasms (NEN) from a gastroenteropancreatic (GEP) or unknown origin. They show that the association of platinum-etoposide chemotherapy with immunotherapy (Nivolumab) allows “interesting” activity in first-line NENs.

That is the first phase II trial testing this combination for G3-NEN. Therefore, I would like to congratulate the authors to have leading this interventional study in a rare disease. I made some comments below, which I hope, could help to improve the manuscript.

Major comments:

I have three major concerns.

1. Firstly, this phase II study mixed well-differentiated neuroendocrine tumour grade 3 (NET-G3) and poorly-differentiated neuroendocrine carcinoma (NEC-G3), while guidelines recommends to separate these two entities as they have a different carcinogenesis, a different prognosis, and they required different systemic treatments; Platinum-etoposide (PE) chemotherapy is the reference treatment for advanced NEC but it's not the standard of care in first-line for NET-G3. However, authors have described the results according to this distinction allowing readers to better understand the results.

2. As highlighted by authors in the study limitations, NICE-NEC is a non-randomized study, and it is therefore difficult to have a good idea of the value of adding nivolumab to the PE chemotherapy. Authors cited the TOPIC-NEC study, but they can better discuss their results in comparison with this large phase III study performed in Asia (very close results obtained without immunotherapy: in this study (n=170), ORR was 54%, median PFS was 5.6 months and median OS was 12.5 months under PE; and in the presented study (NICE-NEC, including 25 NEC), ORR was 54%, median PFS was 5.7 months and median OS was 13.9 months under PE-nivolumab. Only the current phase II/III randomized study will help to demonstrate the value of adding immunotherapy to PE chemotherapy.

3. According to the hypothesis described in the study protocol and Material and Method Section, I understood that the primary end-point (12-month OS) was not reach ($H_0=50\% \Rightarrow H_1=72\%$; the 12-month OS was finally 54.1%, therefore below the H_1). I therefore suggest to temperate the study interpretation in the Abstract and the Discussion Section (“...showed encouraging activity...”, line 67 page 2 “these promising results”, Line 191 page 6 “...which surpassed expected outcomes”: is it true?).

Minor comments:

- Page 3, line 98: please add the reference after “...ARID1A mutations”.
- Page 5, line 149. The results presented (worse ORR/PFS for patients with colorectal NECs) are consistent with those presented by H Sorbye's group (J Neuroendocrinol 2023 Apr;35(4):e13256). This could be discussed (more an effect of PE chemotherapy than immunotherapy?).

- Figure 2A and 2B: the results presented in figure 2 are not consistent with those presented in the text. The words "best" and "worse" at the top of the figures should be inverted.
- Table 1. Please add two columns to describe the study population according to differentiation (NET-G3 and NEC); see the first major comment.
- Table 1. In the protocol and Mat/Met section, the inclusion criteria are for NEN from GEP and unknown origin. However, one patient with a prostate cancer and another one with "inguinal" origin were included. The patient with "inguinal" NEC could be re-classified as an unknown primary with lymph node metastases (if a Merkel cell carcinoma was excluded), but the patient with a prostate cancer should have been excluded from the study. Please explain.
- Table 1. Is the patient with a stage 1 NEN-G3 truly unresectable/advanced NENs as required in the inclusion criteria?

Reviewer #4 (Remarks to the Author): with expertise in neuroendocrine tumors, cancer immunotherapy

Riesco-Martinez and al report on the first trial of immuno-chemotherapy in patients with GEP/ unknown primary high grade neuroendocrine neoplasms.

The manuscript is suitable for publication in Nature Communications given that this is the first trial using this treatment modality in non-pulmonary high grade NENs and in view of encouraging activity observed (high objective response rate, encouraging 1-and 2-year landmark survival) in a patient population of unmet medical need.

Comments:

- The study population is restricted to high grade GEP and unknown primary NENs; table 1 lists however one patient with prostate as the primary site and one patient "inguinal" (?inguinal metastasis involved by high grade NEC; potentially a nodal metastasis of a merkel cell ca with a regressed primary?); both patients should be removed from the data analysis if the protocol specifies GEP and unknown primary as the study population
- Page 5 line 149 should be Figure 1c instead of Figure 1e
- The authors elaborate in the discussion section on the proposed models that could account for an increased efficacy of chemo-immunotherapy compared to anti-PD-1/PD-L1

monotherapy (immunogenic cell death; elimination of immune suppressive cell populations etc); they may also want to reflect on the immediate cytotoxic effect of chemotherapy in highly proliferative malignancies that may stop/slow down tumour growth to allow enough time for anti-PD-1/PD-L1 blockade to induce/re-invigorate a productive anti-tumour immune response without any immuno-modulatory effects

NCOMMS-24-08559

Title: *"Phase II multicenter study of nivolumab plus platinum-doublet chemotherapy in treatment-naive patients with advanced grade 3 Neuroendocrine Neoplasms of gastroenteropancreatic or unknown origin: the NICE-NEC trial (GETNE-T1913)".*

Authors: Riesco-Martinez MC et al.

REVIEWER COMMENTS

REVIEWER #1 (Remarks to the Author): with expertise in neuroendocrine tumors, therapy

The authors are to be commended for this first report of combination chemoimmunotherapy in patients with advanced G3 NENs, and this manuscript is yet another testament to the capabilities of the GETNE group and their exceptional spirit of collaboration to address critical questions in our field. The data from this study are absolutely essential for the neuroendocrine field and should be disseminated through this publication.

We are very grateful to Reviewer #1 for his positive and encouraging comments on our study and our research network activities, and we have addressed the raised issues as detailed in the following point-by-point reply.

There are a few challenges with this study design and results that ought to be discussed in the spirit of maximum transparency, and I would encourage the authors to revise accordingly:

- 1. The primary endpoint was 12-month landmark PFS, and the null hypothesis was not excluded. The authors' conclusion that further study is necessary is reasonable, but explicitly offering this context is appropriate.**

We would first like to clarify that the primary endpoint of the study was the 12-month OS rate (rather than PFS) as described in the Material and Methods section of the manuscript. But following the reviewer's recommendation, we have specified the null and alternative hypothesis for the primary endpoint (12-month OS) in the material and methods section of the abstract, and explicitly stated both in the abstract, results and conclusions sections of the manuscript that the primary endpoint of the trial was not met and thus, the null hypothesis was not rejected.

ABSTRACT

“.....The primary endpoint was the 12-month overall survival (OS) rate (H_0 50%, H_1 72%, β 80%, α 5%).Median PFS was 5.7 months (95%CI: 5.1-9) and median OS 13.9 months (95%CI: 8.3-NR), with a 12-month OS rate of 54.1% (95%CI: 40.2-72.8), over the 50% threshold specified in the null hypothesis, but with a CI lower. Therefore, the null hypothesis could not be rejected. However, 37.6% of patients were long-term survivors (>2 years). Safety profile was consistent with previous reports

RESULTS

“With a median follow-up of 29.8 months (range: 22.2-37.5+) in alive patients, the median OS was 13.9 months (95% CI: 8.3-NR) and the OS rates at 6, 12 and 24 months were 78.4% (95% CI: 66.2-92.8), 54.1% (95% CI: 40.2-72.8) and 37.6% (95% CI: 24.8-57.1), respectively (Figure 1a). The primary endpoint (12-m OS rate) was thus not met....”

CONCLUSIONS

“In conclusion, this study assessed for the first time the addition of ICIs to platinum-based chemotherapy in chemo-naïve patients with advanced or metastatic G3 NENs of GEP or UK origin. Although the primary endpoint (12-month OS rate) was not met, the high proportion of long-term survivors (37.6% OS rate at 2 years) is encouraging in the context of a highly aggressive disease. These findings justify further exploration in larger, randomized trials to more accurately assess the risk-benefit balance of combining immunotherapy with platinum-based chemotherapy as a first-line treatment for patients with high-grade GEP-NENs. Translational studies will also be crucial to identify predictive biomarkers to improve the selection of patients who may benefit the most from this therapeutic strategy.”

- 2. Given the established relationship between Ki-67 and survival, comparisons between populations of patients with different proportions of patients with NETs (with generally lower Ki-67s) is fraught. Explicit accounting for this in both study design and results would be helpful to readers so they can contextualize and interpret the results.**

We acknowledge the inherent limitations of single-arm, non-randomized trials to provide definitive conclusions of improved efficacy, particularly in the context of a heterogeneous patient population such as high-grade neuroendocrine neoplasms. We agree with the reviewer that the close relationship between tumor differentiation and proliferation index (Ki-67), and between Ki-67 and survival, makes comparisons with other studies that may have included different proportions of NETs not reliable. Although exploratory analyses were performed to assess the potential influence of tumor differentiation and Ki-67 index on efficacy, the study was not designed nor powered for such comparisons, as has been clarified in the Material and Methods section (Study endpoints and procedures subsection, lines 394-396) as follows:

“.... Pre-planned subgroup analyses were performed to explore potential signs of differential efficacy by most relevant clinical features, although the study was not powered for formal comparisons; therefore, all findings are exploratory in nature.”

Nevertheless, we believe that the greater strength of our study is the high proportion of patients with prolonged survival, particularly considering that most long-term responders were PD-NECs. Indirect cross-trial comparisons, despite their limitations, are a commonly accepted way to contextualize results in the discussion section and to generate hypothesis that may be subsequently tested in randomized trials. In our opinion, most investigators and scientists are aware of these limitations and will be able to adequately interpret this data. However, to avoid misinterpretations and help the reader contextualize the results, we have included a clarification in the second paragraph of the discussion as follows (lines 252-255):

“...In this context, the results of our study combining ICI with first-line platinum-based chemotherapy are encouraging. However, it should be noted that different proportions of relevant prognostic features in study populations, particularly regarding tumor differentiation, Ki-67 index or primary tumor site, limit comparisons with benchmark studies.”

We would also like to point out that we had already acknowledged these caveats in the submitted version of the manuscript, in the paragraph on the limitations of the study of the discussion section (lines 323-334):

“The main limitation of this trial is the fact that it was a non-randomized study. The lack of a parallel control group restricts the ability to directly compare the treatment outcomes with a standard arm. In addition, the study sample size limits the statistical power to conduct exploratory clinical or molecular analysis to identify subgroups of patients that may obtain a greater benefit, particularly relevant due to the heterogeneity of the study population in terms of primary tumor site and tumor differentiation. Subgroup analyses were not pre-planned and lack statistical power for formal comparisons. These limitations highlight the need for larger, controlled studies to validate our findings and further investigate the added value of combining an ICI with standard chemotherapy. Addressing these limitations, the S2101 (NCT05058651) phase II/III trial, currently underway, aims to enroll 189 patients with extrapulmonary NECs. It will specifically evaluate the impact of adding atezolizumab to standard chemotherapy (carboplatin/cisplatin plus etoposide) compared to chemotherapy alone. This study will help to determine the potential added value of combining ICIs with standard first-line chemotherapy in advanced extrapulmonary G3 NENs.”

- 3. Accordingly, Supplementary Table 1 may merit inclusion in the primary manuscript, rather than relegation to the supplement. Most of the durable responses were in**

patients with PD-NEC that was TMB low and PD-L1 negative. This is exactly the population of patients one would hope to benefit with this combination approach, and I would submit that this may be the most exciting observation of the manuscript.

We agree with the reviewer in that one of the most remarkable findings of our study is that most durable responses were observed in patients with PD-NECs that were PD-L1 negative. Following his recommendation, we have included Supplementary Table 1 in the primary manuscript as Table 2.

To further highlight this relevant observation, we have added the following sentences to the discussion (lines 301-305):

“...Regarding tumor differentiation, our observations and evidence from other trials suggest that NECs obtain greater benefit from ICIs than NETs, possibly due to the higher mutational burden and enhanced neoantigen presentation of poorly differentiated NECs. In fact, 4 out of 5 long-term responders were NECs. However, a low proportion of PD-L1 expression was observed in our cohort (2 patients, 5.4%) and all patients with prolonged duration of response were PD-L1 negative. Similar observations were reported in the DUNE trial, where no correlation was found between PD-L1 expression and efficacy²⁹....”

Overall, the work presented in this manuscript is timely and important. I believe that with some additional detail to support transparent interpretation, it is appropriate for publication.

REVIEWER #2 (Remarks to the Author): with expertise in biostatistics, clinical trial study design

General comments:

1. The abstract should clearly state whether the primary endpoint was met. Since the primary endpoint was not met, I don't understand why authors stated "Randomized trials are required to confirm these promising results."

Following the reviewer's request, we have now clearly stated in the abstract that the primary endpoint was not met. The abstract now reads as follows:

"...Median PFS was 5.7 months (95%CI: 5.1-9) and median OS 13.9 months (95%CI: 8.3-NR), with a 12-month OS rate of 54.1% (95%CI: 40.2-72.8) that did not meet the primary endpoint."

Despite this unquestionable fact, however, we believe the results of our study are encouraging as there was a high proportion of long-term survivors, which is not observed when these patients are treated with conventional chemotherapy only (2-year OS rate $\leq 20\%$ (Morizane et al. 2022; DOI: 10.1001/jamaoncol.2022.3395) and is consistent with the mode of action of immunotherapy. This is particularly remarkable considering that the study cohort was mostly composed of patients with multiple adverse prognostic features, such as poor histological differentiation (68%), Ki-67 index $> 55\%$ (66%), stage IV at diagnosis (95%), ≥ 2 metastatic sites (74%), poor performance status (10.8% ECOG 2) and high baseline enolase levels (63%). In addition, our cohort had low prevalence of molecular alterations predictive of response to immunotherapy, such as positive PD-L1 or MSI, and the majority presented liver involvement at study entry, which can co-opt immune tolerance mechanisms to induce immunotherapy resistance.

This "tail" of long-term survivors is a constant finding of many trials with immune-checkpoint inhibitors in a wide range of tumor types including high grade NENs of GEP (DUNE trial from GETNE, Capdevila et al. 2022; DOI: 10.1038/s41467-023-38611-5) or lung origin (Horn et al, doi: doi: 10.1056/NEJMoa1809064; Paz-Ares et al, doi: 10.1016/S0140-6736(19)32222-6), studies that have led to the approval of durvalumab and atezolizumab for the treatment of small cell lung cancer. In this context, we do believe that if our therapeutic approach is capable of improving long-term survival in a subset of patients, this deserves to be further explored in larger randomized trials.

2. The abstract should also include trial design parameters, such as type I error, power, null hypothesis and alternative hypothesis.

The abstract has been modified and now includes the requested trial design parameters (null and alternative hypothesis, type I error and power), as follows:

"...The primary endpoint was the 12-month overall survival (OS) rate (H_0 50%, H_1 72%, β 80%, α 5%)..."

3. The manuscript shows lack of attention to details and care (see specific comments).

3.1. The reporting of Table 1 and the manuscript text are inconsistent.

Table 1 and the manuscript text have been reviewed and corrected to keep consistency. See reviewed Table 1 and Lines 121-130.

3.2. Figure 1c does not appeared to be referenced anywhere. Maybe Line 149 should be 1c.

Figure 1c has been referenced in the text (line 156 of the current version of the manuscript).

3.3. Line 178, Figure 3A clearly show grade 3+ Neutropenia with more than 50%.

We confirm that the frequency of G3 neutropenia is 39.5%. The typographical error in figure 3A has been corrected.

4. Given that this is a clinical trial report, I would prefer authors to report the primary endpoint (OS) first before all other secondary endpoint.

Following the reviewer's recommendation, the primary endpoint has been moved upfront in the efficacy results section (Lines 138-147)

5. The denominator of ORR and DCR percentage calculation should be all patients who started treatment. Removing patients who died prior to disease assessment (Line 138) from the denominator will make the result appears to be better than it is. The ORR and DCR calculation as currently presented is not appropriate.

We fully agree with the reviewer's criteria and in fact, we did consider and included the full dataset in the denominator (N=37) when calculating all ORR and DCR rates. More specifically, patients who died before the first tumor assessment were considered as treatment failures (disease progression) for ORR and DCR calculations. We just explain in the text that 3 patients died due to clinical disease progression before radiological response assessment could be performed (that is, response was not formally evaluated), but indeed they were accounted as failures and included in the denominator for these endpoints. Nevertheless, to further clarify this important issue, we have modified the texts as follows (lines 166-169)

*“Among the 37 patients evaluable for efficacy, ORR was 56.8% and disease control rate (DCR) 83.8% (Figure 2a). Twenty-one patients (56.8%) had partial response (PR), 10 (27.0%) had stable disease (SD), and 3 (7.9%) had progressive disease (PD) as their best response. Three patients who died due to disease progression before the first tumor assessment **were not evaluated for response but were considered as PD and included in the denominator for all ORR and DCR calculations.**”*

6. A lot of subgroup analysis were presented, e.g. ORR by primary location, ORR by Ki67, and ORR by differentiation. It should be clearly noted in the manuscript that all

these are exploratory analysis. These are not powered; therefore, the p-value should be removed. The confidence interval of the point estimate can be presented. Same comments for PFS/OS by subgroups.

We understand and are fully aware that the study was not powered for formal comparison of efficacy among subgroups. These subgroup analyses, however, were pre-planned to explore potential signs of differential efficacy by most relevant clinical features (i.e. tumor site, differentiation and proliferation rate) and generate hypothesis that may be further explored in future trials. However, following the reviewer's recommendation, we have deleted the p values and specifically emphasized the exploratory nature of these analyses in the material and methods section to avoid overinterpretation by the readers (lines 394-396):

“...Pre-planned subgroup analyses were performed to explore potential signs of differential efficacy by most relevant clinical features, although the study was not powered for formal comparisons; therefore, all findings are exploratory in nature.”

Moreover, this caveat had been specifically acknowledged in the discussion section of the submitted version of the manuscript, in the paragraph on the limitations of the study:

*“The main limitation of this trial is the fact that it was a non-randomized study. The lack of a parallel control group restricts the ability to directly compare the treatment outcomes with a standard arm. In addition, **the study sample size limits the statistical power to conduct exploratory clinical or molecular analysis to identify subgroups of patients that may obtain a greater benefit**, particularly relevant due to the heterogeneity of the study population in terms of primary tumor site and tumor differentiation.”*

7. The multivariable Cox model may be over saturated. Given 35 PFS events, I would not incorporate 9 variables (10 degree of freedom) in the model. Same comment for OS. A reduced number of variables should be used.

Following the reviewer's recommendations, and with the aim of reducing the number of variables in the multivariable Cox models for PFS and OS, we have applied the "backward" stepwise selection method. This method has allowed us to refine the model by systematically eliminating the least significant variables one by one until obtaining a set where all variables are significant. Alternatively, in scenarios where such a set could not be achieved, we have ensured the inclusion of at least the last two remaining variables, which inherently carry more significance. This process ensures the optimization of the resulting model and enhances the robustness of our analyses for both PFS and OS endpoints.

As some significant variables dropped out in these reduced multivariable models, we have modified accordingly the Results section, Figures 1B and 1D, and the discussion section (we have deleted one paragraph and references 35, 36 and 37, and renumbered references as required).

8. I find it curious that the median PFS is 5.7 month but the treatment duration is only 3.5 months for platinum-based chemotherapy and 4.4 months for nivolumab while

majority of patients were off-treatment due to disease progression. Can authors explain why the treatment duration is much shorter than PFS while majority of patients were off-therapy due to progression?

The treatment regimen consisted of 3 drugs, 2 cytotoxic drugs (carboplatin and etoposide) that were to be given for a maximum of six 3-week cycles (18 weeks or 4.5 months) and an immune checkpoint inhibitor (nivolumab) that was to be given for up to 2 years. Patients may develop (and often do) drug-specific toxicities that may lead to the de-escalation of treatment intensity by discontinuing one or two of the drugs but continuing with the others, being the most common scenario that the 6 cycles of chemotherapy may not be completed but the patient can continue with maintenance nivolumab until disease progression or completion of the 2 year of treatment in the absence of progression, but eventually progress thereafter. Thus, it is not surprising that these patients may have shorter duration of chemotherapy than the PFS time, but still come off study due to disease progression. Moreover, in some clinical scenarios, treatment may be interrupted due to clinical suspicion of disease progression, and progression may be confirmed by CT scan some weeks after. It should be also clarified that cycles are administered every 3 weeks for chemotherapy and 4 weeks for maintenance nivolumab. Thus, it is not unexpected that disease progression may be documented 2-6 weeks after the last drug administration.

As illustrated below in the swimmer-plot, the great majority of patients came off study shortly after the last administration of study drug, except for a couple of patients that were progression-free at the date of database lock.

9. Can author provide a citation of the “two-sided one arm survival test”? There are many such tests available.

Following the reviewer’s recommendation, a reference for the two-sided one-arm survival test has been provided (reference 38: Brookmeyer R and Crowley, JJ. *A confidence interval for the median survival time. Biometrics.* <https://doi.org/10.2307/2530286> (1982)) has been provided (lines 576-577)).

10. The decision boundary of the primary endpoint should be stated so the readers can judge whether 54% is barely miss the mark.

In the initial hypotheses, it was assumed that the Overall Survival rate at 12 months for patients treated with Chemotherapy in combination with Nivolumab would be at least greater than 50% (H0A: $p > 50\%$), and it was established that Chemotherapy in combination with Nivolumab would be superior to Chemotherapy if a 72% survival rate at 12 months was achieved (H0B: $p > 72\%$). The results showed a 12-month survival rate of 54.1% (95% CI: 40.2-72.8).

Therefore, while the combination therapy did not meet the predefined criteria for superiority over historical controls treated with chemotherapy alone at the specified threshold, since the 95%CI ranges from 40.2% to 72.8%, neither the null hypothesis nor the alternative hypothesis can be definitively ruled out per study design.

The study hypothesis and the primary outcome point estimate and 95%CI has been provided to the reader in the abstract, methods, results and discussion sections as detailed above, with clear statements specifically stating that the study endpoint was not met.

Specific comments:

1. Line 122, the primary site of Inguinal also has 2 which is the same as esophageal , rectal, and small intestine but Inguinal was omitted from the manuscript text. Is there a reason why?

Table 1 contained some errors regarding primary tumor origin (vs site of metastasis) that have been corrected and are now consistent with the manuscript text.

2. Line 122, table 1 only shows 1 unknown primary but the manuscript text states 5. Please reconcile.

As previously mentioned, Table 1 contained some errors regarding primary tumor origin (vs site of metastasis) that have been corrected and are now consistent with the manuscript text.

4. Line 126, table 1 shows 27 (73.0%) elevated GgA but the text states 79.4%. Please reconcile.

The correct value is 73.0% and has been reconciled between table and text. The reason for the discrepancy in percentages was due to the inclusion or not of the patients with missing values (now included).

4. Line 126, enolase reporting is inconsistent between table 1 and the text.

The correct value is 56.8% and has been reconciled between table and text. The reason for the discrepancy in percentages was due to the inclusion or not of the patients with missing values (now included).

5. Line 126, PD-L1 reporting is inconsistent between table 1 and the text.

PD-L1 values have been reconciled between table and text.

6. Line 156, the word “multivariable” should be used, rather than “multivariate”. Same comments to line 375. Please review the manuscript as a whole to correct all instances.

The text has been adapted following the reviewer’s recommendation.

7. Line 170, please remove “trend toward” the point estimate is either statistical significant or not.

Following the reviewer’s recommendation, we have removed the term “trend toward”.

8. Line 374, I don’t understand what it means by “full range intervals”.

Range intervals represent the min-max values. We explained this and removed the *full* adjective in the methods section for further clarity as follows (lines 436-439):

“Response rates were estimated using 95% confidence intervals (CI), or range (minimum to maximum values) intervals.”

REVIEWER #3 (Remarks to the Author): with expertise in neuroendocrine tumors, therapy

The authors here reported an interesting phase II study on grade 3 (G3) neuroendocrine neoplasms (NEN) from a gastroenteropancreatic (GEP) or unknown origin. They show that the association of platinum-etoposide chemotherapy with immunotherapy (Nivolumab) allows “interesting” activity in first-line NENs.

That is the first phase II trial testing this combination for G3-NEN. Therefore, I would like to congratulate the authors to have leading this interventional study in a rare disease. I made some comments below, which I hope, could help to improve the manuscript.

We are very grateful to Reviewer #3 for his/her positive and encouraging comments on our clinical trial and have addressed the raised issues below.

Major comments:

I have three major concerns.

1. Firstly, this phase II study mixed well-differentiated neuroendocrine tumour grade 3 (NET-G3) and poorly-differentiated neuroendocrine carcinoma (NEC-G3), while guidelines recommends to separate these two entities as they have a different carcinogenesis, a different prognosis, and they required different systemic treatments; Platinum-etoposide (PE) chemotherapy is the reference treatment for advanced NEC but it's not the standard of care in first-line for NET-G3. However, authors have described the results according to this distinction allowing readers to better understand the results.

We agree that carcinogenesis, prognosis and therapeutic strategies differ for G3 NETs and NECs. However, by the time of study design, we used the WHO grading classification of 2010, in which all G3 NENs (NETs and NECs) were grouped together. The category of NET G3 was first acknowledged only for pancreatic NENs in the 2017 WHO classification for endocrine tumors, and was not fully acknowledged for all GI NENs until the 2019 WHO classification for GI tumors, once the trial had started accrual. This new classification took some time to be universally incorporated for diagnosis in routine practice, and a greater time-gap elapsed before it became standard practice to treat G3 NETs and NECs differently in the clinic. In addition, it should be noted that morphologic diagnosis of high-grade NENs is challenging, especially when limited pathologic materials are available, as is often the case in patients with metastatic disease, as there are no clear-cut standardized criteria to discriminate between these 2 entities. Thus, there is a great degree (>30%) of interobserver discrepancy or misdiagnosis among pathologists (Tang LH, et al. Am J Surg Pathol. 2016 Sep;40(9):1192-202; Dinu A, et al, J Gastrointestin Liver Dis. 2023 Jun 22;32(2):162-169). For all these reasons, we decided to include all G3 NENs in this trial and conduct exploratory subgroup analysis by tumor differentiation and Ki-67 index to facilitate interpretation and contextualize the results for these specific subgroups.

2. As highlighted by authors in the study limitations, NICE-NEC is a non-randomized study, and it is therefore difficult to have a good idea of the value of adding nivolumab to the PE chemotherapy. Authors cited the TOPIC-NEC study, but they can better discuss their results in comparison with this large phase III study performed in Asia (very close results obtained without immunotherapy: in this study (n=170), ORR was 54%, median PFS was 5.6 months and median OS was 12.5 months under PE; and in the presented study (NICE-NEC, including 25 NEC), ORR was 54%, median PFS was 5.7 months and median OS was 13.9 months under PE-nivolumab. Only the current phase II/III randomized study will help to demonstrate the value of adding immunotherapy to PE chemotherapy.

We agree with the reviewer that due to the limited sample size and lack of randomization it is not possible to determine the incremental benefit of adding an ICI to standard combination chemotherapy, and thus, results are only hypothesis-generating rather than practice-changing. In fact, our study was negative for the primary endpoint (1-year OS of 72%) as has been clearly stated in the abstract, results and discussion sections of the manuscript. We have also extensively provided in the discussion section data from both western (NORDIC NEC, ECOG ACRIN EA2142) and Asian studies (TOPIC NEC from Japan and a Chinese trial), the latter being more favourable albeit less comparable to our mostly Caucasian population.

Despite this unquestionable fact, however, we believe the results of our study are encouraging as there was a high proportion of long-term survivors, which is not generally observed when these patients are treated with conventional chemotherapy only, including the mentioned TOPIC-NEC trial (2-year OS rate < 20% versus 37.6% observed in our trial). This is particularly remarkable considering that the study cohort was mostly composed of patients with multiple adverse prognostic features, such as poor histological differentiation (68%), Ki-67 index > 55% (66%), stage IV at diagnosis (95%), ≥ 2 metastatic sites (74%), poor performance status (10.8% ECOG 2) and high baseline enolase levels (63%). In addition, our cohort had low prevalence of molecular alterations predictive of response to immunotherapy, such as positive PD-L1 or MSI, and the majority presented liver involvement at study entry, which can co-opt immune tolerance mechanisms to induce immunotherapy resistance.

This “tail” of long-term survivors is consistent with the mode of action of immunotherapy and a constant finding of many trials with immune-checkpoint inhibitors in a wide range of tumor types, including high grade NENs of GEP (DUNE trial from GETNE, Capdevila et al. 2022; DOI: 10.1038/s41467-023-38611-5) or lung origin (Horn et al, doi: doi: 10.1056/NEJMoa1809064; Paz-Ares et al, doi: 10.1016/S0140-6736(19)32222-6), studies that led to the approval of durvalumab and atezolizumab for the treatment of small cell lung cancer. In this context, we do believe that this therapeutic strategy deserves to be further explored in larger randomized trials.

We believe all these statements are fully aligned with the reviewer’s comments, as can be observed in the following selected extracts of our manuscript:

Abstract:

“...The primary endpoint was the 12-month overall survival (OS) rate (H_0 50%, H_1 72%, β 80%, α 5%)....

.....Median PFS was 5.7 months (95%CI: 5.1-9) and median OS 13.9 months (95%CI: 8.3-NR), with a 12-month OS rate of 54.1% (95%CI: 40.2-72.8) **that did not meet the primary endpoint**. However, 37.6% of patients were long-term survivors (>2 years). Safety profile was consistent with previous reports. There was one treatment-related death.

Nivolumab plus platinum-based chemotherapy was associated with prolonged survival in over one third of chemo-naïve patients with G3 GEP-NENs, with a manageable safety profile. **Randomized trials are required to confirm these encouraging results.**"

Results

Treatment efficacy

With a median follow-up of 29.8 months (range: 22.2-37.5+) in alive patients, the median OS was 13.9 months (95% CI: 8.3-NR) and the OS rates at 6, 12 and 24 months were 78.4% (95% CI: 66.2-92.8), 54.1% (95% CI: 40.2-72.8) and 37.6% (95% CI: 24.8-57.1), respectively (Figure 1a). **The primary endpoint (12-m OS rate) was thus not met.**

Discussion:

This is to our knowledge the first prospective trial to evaluate the combination of an ICI and standard first-line chemotherapy in high-grade GEP NENs.....Although **the primary endpoint of this trial was not met (1-year OS rate of 72%)**, the high proportion of long-term survivors is in our opinion particularly encouraging....

.... Cisplatin or carboplatin in combination with etoposide are the most widely used regimens, that yield ORR of 31%, DCR of 64%, and a median survival of 11 months, as reported in one of the benchmark studies, the NORDIC NEC ¹⁹. More recently, **two randomized Asian trials** that compared the combination of cisplatin and etoposide with cisplatin and irinotecan, showed similar efficacy for both regimens, with somewhat higher ORR (53-63%) than those reported in the NORDIC NEC study, but no improvement in OS (10.2-12.5 months) ^{20,21} and a low rate of long-term survivors (2-year OS rate \leq 20%). Another recent randomized trial in a Western population (ECOG ACRIN EA2142) ... reported an ORR of 22%, a median PFS of 5.4 months, and a median OS of 10.6 months for the carboplatin and etoposide combination ²²

In conclusion, this study assessed for the first time the addition of ICIs to platinum-based chemotherapy in chemo-naïve patients with advanced or metastatic G3 NENs of GEP or UK origin. Although **the primary endpoint (12-month OS rate) was not met**, the high proportion of long-term survivors (37.6% OS rate at 2 years) is encouraging in the context of a highly aggressive disease. These findings justify **further exploration in larger, randomized trials** to more accurately assess the risk-benefit balance of combining immunotherapy with platinum-based chemotherapy as a first-line treatment for patients with high-grade GEP-NENs. Translational studies will also be crucial to identify predictive biomarkers to improve the selection of patients who may benefit the most from this therapeutic strategy.

3. According to the hypothesis described in the study protocol and Material and Method Section, I understood that the primary end-point (12-month OS) was not reach (H0=50% => H1=72%; the 12-month OS was finally 54.1%, therefore below the H1). I therefore suggest to temperate the study interpretation in the Abstract and the Discussion

Section (“...showed encouraging activity...”, line 67 page 2 “these promising results”, Line 191 page 6 “...which surpassed expected outcomes”: is it true?).

Following the reviewer’s recommendation, we have tried to temperate the study interpretation through the following actions: 1) we have included in the abstract the main trial design parameters (null and alternative hypothesis, type I error and power), as well as a clear statement indicating that the study endpoint was not met; 2) we have also deleted the statement “...which seem to surpass expected outcomes reported with standard chemotherapy alone” from the second sentence of the discussion; and 3) we have also included explicit statements in the results section and in the discussion (first paragraph summarizing main study results, and last concluding paragraph) indicating that the study endpoint was not met. We have also discussed, however, why we believe that nevertheless the results of the study are interesting enough to deserve further exploration in randomized clinical trials (see comments and manuscript extracts in the answer to question #2).

Minor comments:

- **Page 3, line 98: please add the reference after “...ARID1A mutations”.**

As requested, reference 8 has been added (*Lu M, Zhang P, Zhang Y, et al. Efficacy, safety, and biomarkers of toripalimab in patients with recurrent or metastatic neuroendocrine neoplasms: a multiple-center phase Ib trial. Clin Cancer Res. <https://doi.org/10.1158/1078-0432.CCR-19-4000> (2020)*).

- **Page 5, line 149. The results presented (worse ORR/PFS for patients with colorectal NECs) are consistent with those presented by H Sorbye's group (J Neuroendocrinol 2023 Apr;35(4):e13256). This could be discussed (more an effect of PE chemotherapy than immunotherapy?).**

We have added the following sentence in the discussion section to point out this observation (lines 284-288):

“Conversely, colorectal NENs showed limited benefit from the combination of ICI with chemotherapy, with particularly poor outcomes in line with those reported by Sorbye et al for first-line platinum/etoposide treatment for this subgroup of patients³⁰.”

- **Figure 2A and 2B: the results presented in figure 2 are not consistent with those presented in the text. The words "best" and "worse" at the top of the figures should be inverted.**

Results in Figure 2 and text have been reconciled. The figure has been adapted accordingly.

- **Table 1. Please add two columns to describe the study population according to differentiation (NET-G3 and NEC); see the first major comment.**

Following the reviewer's request, two columns have been added to Table 1 to provide characteristics of the study population for these subgroups.

- **Table 1. In the protocol and Mat/Met section, the inclusion criteria are for NEN from GEP and unknown origin. However, one patient with a prostate cancer and another one with "inguinal" origin were included. The patient with "inguinal" NEC could be re-classified as an unknown primary with lymph node metastases (if a Merkel cell carcinoma was excluded), but the patient with a prostate cancer should have been excluded from the study. Please explain.**

Table 1 contained some typos/errors regarding primary tumor origin (vs site of metastasis) that have been corrected. The prostatic was a pancreatic primary. The inguinal NEC was indeed an unknown primary with extensive pelvic lymph node metastasis (Merkel cell carcinoma excluded).

- **Table 1. Is the patient with a stage 1 NEN-G3 truly unresectable/advanced NENs as required in the inclusion criteria?**

Yes, as this refers to tumor stage at diagnosis, not at study entry. The referred patient had a stage I G3 colorectal NEC at initial diagnosis that was surgically resected with curative intent, and eventually developed liver and peritoneal metastatic disease and was then included in the trial.

REVIEWER #4 (Remarks to the Author): with expertise in neuroendocrine tumors, cancer immunotherapy

Riesco-Martinez and al report on the first trial of immuno-chemotherapy in patients with GEP/ unknown primary high grade neuroendocrine neoplasms. The manuscript is suitable for publication in Nature Communications given that this is the first trial using this treatment modality in non-pulmonary high grade NENs and in view of encouraging activity observed (high objective response rate, encouraging 1-and 2-year landmark survival) in a patient population of unmet medical need.

We are very grateful to Reviewer #4 for his/her positive and encouraging comments on our clinical trial and have addressed the raised issues below.

Comments:

• The study population is restricted to high grade GEP and unknown primary NENs; table 1 lists however one patient with prostate as the primary site and one patient “inguinal” (?inguinal metastasis involved by high grade NEC; potentially a nodal metastasis of a merkel cell ca with a regressed primary?); both patients should be removed from the data analysis if the protocol specifies GEP and unknown primary as the study population

Table 1 contained some typos/errors regarding primary tumor origin (vs site of metastasis) that have been corrected. The prostatic was a pancreatic primary. The inguinal NEC was indeed an unknown primary with extensive pelvic lymph node metastasis (Merkel cell carcinoma excluded). Thus, all patients are correctly included in the analysis.

• Page 5 line 149 should be Figure 1c instead of Figure 1e

This typographical error has been corrected in the text.

• The authors elaborate in the discussion section on the proposed models that could account for an increased efficacy of chemo-immunotherapy compared to anti-PD-1/PD-L1 monotherapy (immunogenic cell death; elimination of immune suppressive cell populations etc); they may also want to reflect on the immediate cytotoxic effect of chemotherapy in highly proliferative malignancies that may stop/slow down tumour growth to allow enough time for anti-PD-1/PD-L1 blockade to induce/re-invigorate a productive anti-tumour immune response without any immuno-modulatory effects

We agree with the reviewer's consideration regarding the important role of chemotherapy in the combination regimen to induce a rapid cytotoxic effect in these rapidly progressing malignancies and thereby enabling enough time for immune checkpoint inhibitors to induce/re-invigorate a productive anti-tumour immune response. Following his/her recommendation, we have completed the alluded discussion section as follows (lines 280-282):

“Chemotherapy may improve the immunological effects of ICIs as it induces an immunogenic cell death, reduces regulatory T-cell activity and induces PD-L1 expression, and this synergy has in fact been demonstrated in the clinical setting^{6,7}. Moreover, chemotherapy has short-term cytotoxic effects that may slow down tumor growth and allow enough time for ICI to induce/re-invigorate an effective antitumor immune response. Results are particularly encouraging in G3 GEP NENs of non-colorectal origin (ORR 70.8%), although small numbers and lack of control preclude definitive conclusions.”

REVIEWERS' COMMENTS

Reviewer #1 (Remarks to the Author):

Comments have been appropriately addressed.

Reviewer #2 (Remarks to the Author):

Thank you for addressing my statistical comments. Majority of my comments have been addressed to satisfaction except one.

The reference provided for the “two-sided one arm survival test” does not appear to be correct. Brookmeyer and Crowley’s paper discussed the confidence interval for the median survival time. The primary endpoint of this trial is OS at 12 months; therefore, the test has to be one that tests the survival proportion at a fixed-time. Please provide a correct reference for the statistical design.

Reviewer #3 (Remarks to the Author):

Thank you to give me again the opportunity to review this article NCOMMS-24-08559A entitled “Phase II multicenter study of nivolumab plus platinum-doublet chemotherapy in treatment-naive patients with advanced grade 3 Neuroendocrine Neoplasms of gastroenteropancreatic or unknown origin: the NICE-NEC trial (GETNE-T1913)”.

I congratulate the authors who have addressed all the comments made by the 4 Reviewers. They have acknowledged that their study is negative according to their hypothesis.

Comments:

1. Even with the corrections made by authors according to the request of the Reviewer #2, the presentation of the univariate and multivariate (Cox model) analysis remains unclear in Text and Figure 1. Please detail, at least in Methodology Section and in Supplementary data, which variables were tested in univariate analysis and which variables were put in the Cox model, and therefore why you present different variables between PFS and OS (Figure 1).
2. I think you have all data to calculate the GI-NEC score (Ki67, ALP, LDH, liver met, PS)

described by Lamarca et al. How is this variable on PFS and OS after univariate and multivariate analysis?

Reviewer #4 (Remarks to the Author):

All my comments raised have fully been addressed

REVIEWERS' COMMENTS

REVIEWER #1 (Remarks to the Author)

Comments have been appropriately addressed.

Thank you very much for your contribution to improve our work.

REVIEWER #2 (Remarks to the Author)

Thank you for addressing my statistical comments. Majority of my comments have been addressed to satisfaction except one.

The reference provided for the “two-sided one arm survival test” does not appear to be correct. Brookmeyer and Crowley’s paper discussed the confidence interval for the median survival time. The primary endpoint of this trial is OS at 12 months; therefore, the test has to be one that tests the survival proportion at a fixed-time. Please provide a correct reference for the statistical design.

Reference 35 has been modified as requested. We have now included the link to the software used to design the study to test the survival proportion at a fixed time (<https://stattools.crab.org/Calculators/oneNonParametricSurvival.htm>).

REVIEWER #3 (Remarks to the Author)

Thank you to give me again the opportunity to review this article NCOMMS-24-08559A entitled “Phase II multicenter study of nivolumab plus platinum-doublet chemotherapy in treatment-naive patients with advanced grade 3 Neuroendocrine Neoplasms of gastroenteropancreatic or unknown origin: the NICE-NEC trial (GETNE-T1913)”.

I congratulate the authors who have addressed all the comments made by the 4 Reviewers.

They have acknowledged that their study is negative according to their hypothesis.

Comments:

1. Even with the corrections made by authors according to the request of the Reviewer #2, the presentation of the univariate and multivariate (Cox model) analysis remains unclear in Text and Figure 1. Please detail, at least in Methodology Section and in Supplementary data, which variables were tested in univariate analysis and which variables were put in the Cox model, and therefore why you present different variables between PFS and OS (Figure 1).

Variables tested in univariable and multivariable analysis have been described in the Methods section as follows (see *statistical analysis* subsection of Methods):

“.....Variables assessed in univariable subgroup analysis to explore potential signs of differential efficacy by most relevant clinical features included the following: primary tumor site, tumor differentiation, Ki-67 index, ECOG performance status, sex, age, chromogranin A, neurospecific enolase and LDH. Multivariable regression models also assessed these variables to analyze their potential relationship with efficacy endpoints. To obtain the multivariable models, we employed the 'backward' stepwise selection method, systematically eliminating non-significant exploratory variables until a significant set was achieved. In instances where a significant set could not be reached, the last two remaining variables were included for analysis (see supplementary information for further details)....”

Further details of these analyses have been provided in the Supplementary data as requested (Supplementary figures 2 and 4).

2. I think you have all data to calculate the GI-NEC score (Ki67, ALP, LDH, liver met, PS) described by Lamarca et al. How is this variable on PFS and OS after univariate and multivariate analysis?

Following the reviewer’s suggestions we explored *post hoc* the Lamarca et al. GI-NEC score to show its distribution and correlation with survival outcomes. No significant differences were found in our cohort.

However, we would rather not include this analysis in our manuscript as this is not a standard score used in routine clinical practice, our study was not adequately powered for it, and it was not a pre-planned analysis. Therefore, it may be misleading or lead to controversial conclusions.

Analysis of GI-NEC score in NICE-NEC:

Table. Distribution of patients by GI-NEC score.

GI-NEC Score	N 37
0	3 (8.1%)
1	2 (5.4%)
2	16 (43.2%)
3	7 (18.9%)
4	9 (24.3%)

The multivariable hazard ratio for GI-NEC score 3-6 vs 0-2 was 1.91 (95% CI: 0.92 - 3.96) and 2.49 (95% CI: 0.86 - 7.19) for PFS and OS, respectively. No significant differences were found in univariable analysis.

REVIEWER #4 (Remarks to the Author)

All my comments raised have fully been addressed.

Thank you very much for your contribution to improve our work.